

# Evidence of intense climate variation and reduced ENSO
# activity from δ$^{18}$O of *Tridacna* 3700 years ago
Yue Hu [a,b], Xiaoming Sun [a,b,c,d*], Hai Cheng [e,f], Hong Yan [g, h, i*]
*[a] School of Marine Sciences, Sun Yat-sen University, Guangzhou 510006, China*
*[b] Guangdong Provincial Key Laboratory of Marine Resources and Coastal Engineering,*
*Guangzhou 510275, China*
*[c] School of Earth Sciences and Engineering, Sun Yat-sen University, Guangzhou 510275, China*
*[d] Southern Marine Science and Engineering Guangdong Laboratory (Zhuhai), Zhuhai 519000,*
*China*
*[e] Institute of Global Environmental Change, Xi'an Jiaotong University, Xi'an 710054, China*
*[f] Department of Earth Sciences, University of Minnesota, Minneapolis, Minnesota 55455, USA*
*[g] State Key Laboratory of Loess and Quaternary Geology, Institute of Earth Environment, Chinese*
*Academy of Sciences, Xi'an 710061, China*
*[h] CAS Center for Excellence in Quaternary Science and Global Change, Xi'an 710061, China*
*[i] OCCES, Qingdao National Laboratory for Marine Science and Technology, Qingdao 266061,*
*China*
*Corresponding authors: eessxm@mail.sysu.edu.cn; yanhong@ieecas.cn
## Abstract
*Tridacna* is the largest marine bivalves in the tropical ocean, and its carbonate shell can shed
light on high-resolution paleoclimate reconstruction. In this contribution, δ$^{18}$O$_{shell}$ was used to
estimate the climatic variation in the Xisha Islands of the South China Sea. We first evaluate the
sea surface temperature (SST) and sea surface salinity (SSS) influence on modern rehandled
monthly (r-monthly) resolution *Tridacna gigas* δ$^{18}$O$_{shell}$. The obtained results reveal that δ$^{18}$O$_{shell}$
seasonal variation is mainly controlled by SST and appear insensitive to local SSS change. Thus,
the δ$^{18}$O of *Tridacna* shells can be roughly used as a proxy of the local SST: a 1 ‰ δ$^{18}$O$_{shell}$ change

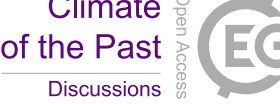

is roughly equal to 4.41 °C of SST. R-monthly $\delta^{18}O$ of a 40-year *Tridacna squamosa* ($3673 \pm 28$
BP) from the North Reef of Xisha Islands was analyzed and compared with the modern specimen.
The difference between the average $\delta^{18}O$ of fossil *Tridacna* shell ($\delta^{18}O$ = -1.34 ‰) and modern
*Tridacna* specimen ($\delta^{18}O$ = -1.15 ‰) probably implies a warm climate with roughly 0.84°C higher
in 3700 years ago. The seasonal variation in 3700 years ago was slightly decreased compared with
that suggested by the instrument data, and the switching between warm and cold-seasons was
rapid. Higher amplitude in r-monthly and r-annual reconstructed SST anomalies implies an
enhanced climate variability in this past warm period. Investigation of the El Ninõ-Southern
Oscillation (ENSO) variation (based on the reconstructed SST series) indicates a reduced ENSO
frequency but more extreme El Ninõ events in 3700 years ago.

***Key words***: *Tridacna*; $\delta^{18}O$; South China Sea; Seasonal variation; Climate variation; ENSO
activity

## 1 Introduction

Carbonate skeleton of marine organisms, such as corals, foraminifers, mollusks, have been

widely used to reconstruct environmental variation (Aharon, 1983; Batenburg et al., 2011; Ourbak
et al., 2006; Schöne et al., 2005; Wanamaker et al., 2011; Yoshimura et al., 2016; Yu et al., 2005).
Due to their high sensitivity to the surrounding environment and the ability to preserve of
high-resolution physicochemical variations in their skeleton, these marine biogenic carbonates can
shed light on the past climate dynamics. *Tridacna* species, as the largest bivalves and usually live
in tropical coral reefs, have received increasing scientific attention in the recent decades (Pätzold
et al., 1991; Watanabe et al., 1999; Watanabe et al., 2004; Elliot et al., 2009; Ayling et al., 2015).
This is because these bivalves and their shells have many favorable properties for recording local
environmental changes: they have dense and well-preserved aragonite shells, fast growth rates (up
to 1 cm/yr) with clear annual growth lines, and with longevity from several decades to a few
centuries. These advantages make *Tridacna* an ideal material for high-resolution reconstruction of
interannual, seasonal or even sub-seasonal climatic variations.

Previous studies indicated that *Tridacna* species precipitate their shells with the oxygen



isotopic ($\delta^{18}$O) equilibrium with seawater (Aharon, 1991; Aharon and Chappell, 1986; Pätzold et
al., 1991; Romanek and Grossman, 1989; Watanabe et al., 1999), and the influence of ontogenic
reduction on the *Tridacna* $\delta^{18}$O is negligible (Welsh et al., 2011). These studies implied that
$\delta^{18}$O$_{shell}$ can be used to reconstruct the late Quaternary Sea-level and climatic changes. Indeed,
$\delta^{18}$O of marine biogenic carbonates are not only influenced by sea surface temperature (SST) but
also by surrounding seawater $\delta^{18}$O. Meanwhile, seawater $\delta^{18}$O have a close correlation with sea
surface salinity (SSS), which is affected by tropical evaporation and precipitation balance.
Nonetheless, the SST and SSS influence on $\delta^{18}$O$_{shell}$ is uncertainties due to the distinct variation of
temperature and salinity in different area. For example, $\delta^{18}$O$_{shell}$ of the *Tridacna* from
southwestern Japan could be directly used as a proxy of SST (Yamanashi et al., 2016), while
$\delta^{18}$O$_{shell}$ of Indonesian *Tridacna* were interpreted to be contributed 71.4 % by SST and 28.6 % by
SSS (Arias-Ruiz et al., 2017). Thus, local calibration from modern *Tridacna* is important to
determine the relationship of $\delta^{18}$O$_{shell}$, SST and SSS.

Climatic variation in the Meghalayan (began at 4200 BP in late Holocene) has significant

impacts on human society and ecosystem development. However, the early Meghalayan climatic
conditions in SE Asia around the South China Sea still remain poorly understood. Shi (1994)
reviewed the data from various sources (like ice core, inland lakes, paleosols in loess and eolian
sands, sea level fluctuations, palynological and botanical studies) in China, indicating the early
Maghalayan was involved in Holocene Megathermal period (8 to 3 ka BP). Sediments in the
South China Sea also implied the temperature may have been relatively higher in the early of
Meghalayan than present (Ouyang et al., 2016). However, those studies are low- resolution, the
high-resolution records under interannual climate variation are rare. With global warming and
many climatic disasters occur nowadays, the climatic conditions in the early Meghalayan could
serve as an analogue to the modern problems, and have received increasing scientific attention
(Schirrmacher et al., 2019; Scuderi et al., 2019; Toth and Aronson, 2019; Zhang et al., 2018).
High-resolution isotopic geochemical data on the *Tridacna* in this period become an insight into
the climatic variations, including extreme ones.

Furthermore, the El Ninô-Southern Oscillation (ENSO) is widely accepted to be a main

trigger for interannual climatic variability in the Pacific Ocean. Previous studies suggested that the

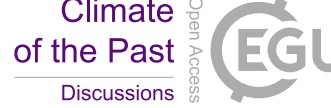

impacts of ENSO activity would not be limited to the tropical area, but also on the global
atmospheric circulation through heating-up of the tropical atmosphere (Cane, 2005). A
fragmentary understanding of the ENSO dynamics causes the uncertainties to predict current or
future variation. Many published models of ENSO behavior (on the average climate and
background of the tropical Pacific) were constructed with low-resolution proxy data (Clement et
al., 1999), so it seems seasonal or monthly data are important to examine the precise variation in
ENSO activity. Recent studies on the late Holocene ENSO evolution yielded controversial
findings: Coral records from the tropical Christmas Island showed a reduced ENSO variability
around the late Holocene (McGregor et al., 2013; Woodroffe et al., 2003), yet some other studies
indicate strengthening ENSO activity at 4 to 3 ka BP (Tudhope et al., 2001; Duprey et al., 2014;
Yang et al., 2019). Thus, this further points to the importance of high-resolution isotopic
geochemical data in unraveling the dynamics of ENSO from the local to global scale.

This study aims to evaluate the seasonality, climate variation, and ENSO activity in the

Xisha Islands of the northern South China Sea, based on two high-resolution $\delta^{18}O_{shell}$ profiles of
modern and fossil *Tridacna*. The study area situated in the northwest margin of the West Pacific
Warm Pool (WPWP), and the local climate is widely accepted to be directly responsive to ENSO
activity (Mitsuguchi et al., 2008; Yan et al., 2010). A modern *Tridacna gigas* shell was first to
estimate the extent of environmental control (SST and SSS) on $\delta^{18}O_{shell}$, and a new SST-$\delta^{18}O_{shell}$
linear regression was proposed. Subsequently, a 40-year fossil *Tridacna squamosa* was used to
reconstruct the seasonality and climatic variation, and the obtained results are compared with the
modern species and meteorological observations. Finally, the ENSO activity and extreme El Ninð
events were discussed, using the re-established SST anomalies.

## 2 Materials and methods

### 2.1 Regional setting

The South China Sea is located in the northwest of WPWP (Fig. 1a), and its interannual

climate has a close relation to ENSO activities (Mitsuguchi et al., 2008; Yan et al., 2010). The
Xisha Islands in the northern South China Sea (300 km south of Hainan Island) is substantially
influenced by two contrasting Asian monsoons from opposite directions: The Asian summer



monsoon from the southwest and the Asian winter monsoon from the northeast. These two
monsoons give distinct seasonal SST to the *Tridacna* from the coral reefs of the Xisha Islands.
Our sample (*Tridacna squamosa* A5) was collected in the North Reef (17°05′ N, 111°30′ E),
whilst the modern *Tridacna gigas* sample YX1 (studied previously by Yan (2013)), was acquired
from the Yongxing Island (16°50′ N, 112°50′ E), which is about 90 kilometers away from the
North Reef (Fig. 1a).

Meteorological observations (atmosphere temperature (AT), SST, SSS, rainfalls) are obtained

from the Institute of Meteorology of China in the Xisha Islands since 1958. Due to the minimum
number of YX1 in a year is seven, the time-scale of modern *Tridacna* YX1 is rehandling into
seven points/yr, which indicates a rehandled month (r-month) represents 1.7 actual month. All
meteorological observations and $\delta^{18}O_{shell}$ are using this method to rehandle the time-scale. Figure
1d shows the r-monthly-average time series of AT, SST, SSS, rainfall and their standard deviations
(SD). The mean SST is 27.77 °C, AT show a highly positive correlation with SST (r=0.98), but is
0.7 °C lower. The SST seasonality is 5.33 °C, with the lowest value and highest value occurring in
1$^{st}$ r-month and 4$^{th}$ r-month, respectively. The Xisha Islands are far from the continent river runoff
can hardly influence on SSS. SSS change from 33.25 to 33.81 ‰, and the change is mainly
dominated by rainfall: higher SSS in dry winter and lower SSS in wet summer (Fig. 1e).

The SST data in the North Reef are acquired from NOAA HadISST, a global monthly SST

data with a spatial resolution of 1° × 1° (data grid cell of data includes both the North Reef and the
Yongxing Island) from 1982 to 2017. Ninõ 1 + 2 SST are obtained from NOAA monthly data
between 1982 to 2017 (http://www.cpc.ncep.noaa.gov/data/indices/sstoi.indices).
*2.2 Shell descriptions and sample preparation*

The 29 cm long fossil *Tridacna squamosa* A5 was cut from the umbo to the ventral margin

along the axis of maximum growth (Fig. 1b). A 5 mm-thick slice reveals three different zones (Fig.
1c): the inner layer, outer layer and the hinge. The inner layer is chosen for the analyses because
of its clear growth layer and well-preserved shell. Published data also revealed that the inner layer
$\delta^{18}O$ values were unaffected by different growth rates or ontogeny (Welsh et al., 2011), and could
better reflect actual $\delta^{18}O$ than the inner layer or the hinge (Pätzold et al., 1991; Elliot et al., 2009).

The $^{14}C$ AMS test revealed the fossil *Tridacna gigas* age was 3437 ± 28 BP. For the



marine-reservoir effect, the conventional radiocarbon age was 3673 ± 28 BP using the
Radiocarbon Calibration Program CALIB 7.10 (http://calib.org). Both X-ray diffraction (XRD)
and laser Raman spectrometers results were aragonite, no other substances were found.
*2.3 Stable isotopes*
Stable isotope samples were micromilled perpendicular to the growth layer under the
micro-drill automated system (Micro-Drill New Wave Research, Olympus SZ 61) in the Isotope
Laboratory of Xi'an Jiaotong University, China. Each sample was performed under 1 mm long,
100 μm deep. Four intervals were used according to the growth rates: 100 μm (n = 1 to 268), 150
μm (n = 269 to 481), 200 μm (n = 482 to 657), 300 μm (n = 658 to 765) respectively from adult to
childhood (Fig. 2).
$\delta^{18}$O of *Tridacna* was analyzed in the Isotope Laboratory of Xi'an Jiaotong University, using
the ThermoFinnigan MAT-253 mass spectrometer fitted with a Kiel Carbonate Device IV. All the
results were reported in per mil (‰), relative to the Vienna PeeDee Belemnite (VPDB) standard.
The international standard $TTB_1$ were added to the analyses every 10 to 20 samples to check the
reproducibility. Duplicate measurements of $TTB_1$ standards and samples showed a long-term
reproducibility (1σ) of less than 0.14 ‰ and 0.05 ‰, respectively.
Published data of the modern *Tridacna* gigas shell YX1 were used to investigate the
relationship between *Tridacna* $\delta^{18}$O and local climate (Yan et al., 2013). YX1 was collected from
the Yongxing Island, 120 km ESE of the North Reef (Fig. 1b). Modern *Tridacna* YX1 $\delta^{18}$O
(VPDB) of internal carbonate standard (GBW04405) is of (average) -8.49 ± 0.14 ‰, and the
standards and samples have reproducibility (1σ) of better than 0.08 ‰ and 0.06 ‰, respectively.
The average $\delta^{18}$O (VPDB) TTB1 (A5) is also of -8.49 ± 0.14 ‰, which would minimize deviation
during comparison.
*2.4 Data processing and analyses*
PearsonT3 (Version 2.2, January 2017) was used to test the correlation coefficient. Monthly
insolation was calculated in 100 years by AnalySeries 2.0.8 (Laskar et al., 2004), which contained
the calculated sigmas of conventional radiocarbon age in *Tridacna* (A5) life span. The years of
modern insolation range from 1918 to 2017, and the time-scale of *Tridacna* A5 range from 3722 to
3623 BP. Statistical analyses were performed with software of Origin 2018 and PAST 3.18. Since





the yearly minimum number in $\delta^{18}O_{YX1}$ was seven, thus the isotopic records, climatic data and
insolation data were rehandled to seven points/yr with the AnalySeries 2.0.8 (Schöne and Fiebig,
2009; Wanamaker et al., 2011). This sclerochronologic rehandling would decrease the growth
rates deviation.

## 179   3 Results

*3.1 $\delta^{18}O_{A5}$ record*
Seasonal cycles are distinct in the $\delta^{18}O_{A5}$ profile (Fig. 2), which show the 40 years of which
the *Tridacna* had lived. The $\delta^{18}O_{A5}$ range from -2.07 to -0.14 ‰ (mean -1.35 ‰, n=765). After
rehandling into 7 points/yr, $\delta^{18}O_{A5}$ vary from -1.98 to -0.29 ‰ (mean -1.34 ‰, n=281).
*3.2 Sclerochronology*
From the shell slice section, 40 dark/light couples (each representing one year) can be seen
clearly. Higher $\delta^{18}O_{A5}$ values lie in the short dark increments (transparent), corresponding to the
low temperature and dry seasons. In contrast, lower $\delta^{18}O_{A5}$ values lie in the long light increments
(opaque), corresponding to the high temperatures and wet seasons (Fig. 3a).
Annual growth rates can be calculated with the $\delta^{18}O_{A5}$ seasonal cycles and interval distance
(Fig. 3b). The results show that growth rates were higher when *Tridacna* A5 was young, reaching
5 mm/yr. The growth then slowed down and stabilized to 1-2 mm/yr after the *Tridacna* had grown
mature. Furthermore, daily increments are obvious under the microscope (Fig. 3b). In general,
*Tridacna* A5 grew faster in warm seasons and slower in cold seasons (Fig. 3b).
The SST observation in the Xisha Islands suggested that the 1[st] r-month corresponds to
almost the lowest SST. Thus, the highest $\delta^{18}O$ of each cycle was chosen to be the beginning of a
year. After the data rehandling, the potential deviation in different growth rates can also be
reduced.

## 199   4 Discussion

*4.1 Relation of SST, SSS and $\delta^{18}O$ of modern Tridacna*
Previous studies demonstrated that *Tridacna* is in isotopic equilibrium with the surrounding
seawater (Aharon, 1983; Watanabe et al., 1999), which also holds true for the *Tridacna* in the



South China Sea (Yan et al., 2013). Biogenic carbonate $\delta^{18}O$ values are in linear correlations with
the SST and seawater $\delta^{18}O_{water}$ (Aharon and Chappell, 1986; Pätzold et al., 1991; Romanek and
Grossman, 1989). We adopted the $\delta^{18}O_{shell}$-SST-$\delta^{18}O_{water}$ Eq. (1) of Grossman and Ku (1986),
which is widely used in calculations for tropical aragonite mollusk species. Meanwhile, $\delta^{18}O_{water}$
has a positive relationship to SSS, thus, $\delta^{18}O_{water}$ can be estimated with Eq (2) which is established
through seawater in the northern South China Sea (Hong et al., 1997). We merged Eq (1) and (2)
into $\delta^{18}O_{shell}$-SST-SSS (Eq (3)), and used two approaches to discuss the extent of SST and SSS
influence on $\delta^{18}O_{YX1}$ under different time-scale.
SST (°C) = 21.8 - 4.69 ($\delta^{18}O_{shell}$ - $\delta^{18}O_{water}$)                                                                 (1)
$\delta^{18}O_{water}$ (‰) = 0.23 × SSS − 7.58                                                                                         (2)
SST (°C) = -13.75 − 4.69 × $\delta^{18}O_{shell}$ + 1.08 × SSS                                                           (3)
In the first approach (seasonal time-scale), we hypothesized two conditions: one with
constant SSS but varying SST, and the other with constant SST but varying SSS. Two $\delta^{18}O$
profiles can be calculated: $\delta^{18}O_{SST}$ (under constant SSS) and $\delta^{18}O_{SSS}$ (under constant SST) (Fig.
4a). R-monthly mean values were used to minimize the influence of extreme events. The $\delta^{18}O_{YX1}$,
$\delta^{18}O_{SST,}$ and $\delta^{18}O_{SSS}$ values are of -0.57 to -1.52 ‰, -0.48 to -1.58 ‰, -1.07 to -1.19 ‰,
respectively. The $\delta^{18}O_{SSS}$ variation is only 0.12 ‰, 14 % of the $\delta^{18}O_{YX1}$ variation. The correlation
between $\delta^{18}O_{YX1}$ and $\delta^{18}O_{SST}$ is high (r = 0.91, n = 7; r = 0.78, n = 77), and the two $\delta^{18}O$ profiles
show the same trend. This indicates that $\delta^{18}O_{shell}$ in the Xisha Islands correspond predominantly to
the seasonal SST variation.
In the second approach (based on Eq (1) and (2)), the calculated $\delta^{18}O_{predicted}$ (by using both
actual SST and SSS) were used to compare with $\delta^{18}O_{YX1}$ (Table S1). The $\delta^{18}O_{YX1}$ and $\delta^{18}O_{predicted}$
profiles have nearly the same mean value (1.15 ‰ and 1.14 ‰, respectively) and indicate a
perfect match (r = 0.81, n = 77). This confirms that the local *Tridacna* precipitates its shell in
oxygen isotopic equilibrium. In order to determine whether the SSS variation in different season
affect the predicted SST significantly, we use the actual SSS, constant SSS (mean SSS) and
$\delta^{18}O_{YX1}$ to calculate predicted SST. Two predicted SST values (one calculated with varying SSS
and the other with constant SSS) have high similarity (r = 0.93) (Fig. 4e), and they correspond to
the variation of actual SST. Each of these predicted SST values is well correlated with the actual



SST ($r_{vary}$ = 0.79, $r_{constant}$ = 0.78). This means that the SSS has little influence on the seasonal
$\delta^{18}O_{shell}$ variation. Thus, we can then use $\delta^{18}O_{shell}$ to roughly estimate the seasonal local SST
variation, and establish a new SST-$\delta^{18}O_{shell}$ linear regression: SST (°C) = 22.69 - 4.41 × $\delta^{18}O_{shell}$
(or $\delta^{18}O_{shell}$ (‰) = -0.136 × SST + 2.634). A 1 ‰ change of $\delta^{18}O_{shell}$ is roughly equal to 4.41°C of
SST. Yu (2005) summarized the published $\delta^{18}$O-SST slopes for the *Porites lutea* coral from
different places, and suggested that the slopes range from -0.134 to -0.189, in which our result lies
(-0.136). In addition, corals from Hainan Island revealed a good $\delta^{18}$O vs. SST correlation with a
linear regression slope of -0.137 (Su et al., 2006), very similar to our result. Consequently, it is
reliable to use the new linear regression for reconstructing the past SST with the fossil $\delta^{18}O_{shell}$.
*4.2 Indication of seasonal variation in modern Tridacna*

From both $\delta^{18}O_{YX1}$ (-0.60 to -1.52 ‰) and $\delta^{18}O_{predicted}$ (-0.47 to -1.57 ‰) profiles (Fig. 4b),

clear seasonality in shown with the lowest value occurring in the 1st r-month (cold seasons) and
the highest value in the 4th r-month (warm seasons). Variance in $\delta^{18}O_{YX1}$ seasonality is 0.19 %
shorter than $\delta^{18}O_{predicted}$, which may be due to the different growth rates and equidistance sampling
mode. In each year, the analyzed *Tridacna* grew faster in warmer seasons than in colder seasons,
thus, specimens under equidistance sampling mode would have more samples in the warm seasons.
Fewer points in the cold seasons would decrease the values and lead to lower $\delta^{18}O_{shell}$ in the 1st
r-month, but the higher number of points make $\delta^{18}O_{shell}$ close to $\delta^{18}O_{predicted}$ in the warm seasons
(nearly identical in the 4th r-month. Moreover, throughout the life of the analyzed *Tridacna*, the
$\delta^{18}O_{shell}$ amplitude is more approached to the actual $\delta^{18}O_{predicted}$ under higher number of points
(high growth rates) before it reached maturity. After the *Tridacna* reach maturity, the fewer points
taken in a year yielded a lower amplitude. This can explain the minor discrepancy between
$\delta^{18}O_{shell}$ and $\delta^{18}O_{predicted}$. As a result, $\delta^{18}O_{shell}$ would slightly reduce the actual seasonal variation.
However, the correlation between them is high (r = 0.81, n = 77), and the mean of $\delta^{18}O_{YX1}$
(-1.15 ‰) and $\delta^{18}O_{predicted}$ (-1.14 ‰) values are similar. Therefore, $\delta^{18}O_{shell}$ can also be used to
estimate the actual seasonal variation, with caution to the slightly reduced variation.
*4.3 Reconstructed climate with fossil Tridacna A5 $\delta^{18}$O evidence*

The fossil *Tridacna* lived in 3700 years ago during the early Meghalayan. The 40 $\delta^{18}O_{A5}$

cycles reveal that *Tridacna* A5 had probably lived for at least 40 years. After calculating data into



r-monthly average profiles, the extreme seasonal variation effects were minimized. The mean
$\delta^{18}O_{A5}$ profiles is -1.34 ‰, with the minimum and maximum of -1.66 and 0.66 ‰, respectively
(Fig. 4c). Contrasting to the mean value of YX1 (-1.15 ‰), the lower $\delta^{18}O_{A5}$ mean value may have
reflected the higher temperature in which the *Tridacna* had lived. To translate into SST (without
considering the SSS changes), the temperature was estimated to be roughly 0.84°C higher than
present. This agrees with other lines of evidence that suggested a higher temperature during that
period (Ouyang et al., 2016), which was considered to be a Holocene Megathermal in China (8.5
to 3 ka BP) (Shi et al., 1992).
The average r-monthly seasonal range of this period (1 ‰) is similar to that yielded from
YX1 (0.92 ‰). The standard deviations of $\delta^{18}O_{A5}$ (0.38 ‰, n=281) and $\delta^{18}O_{YX1}$ (0.35 ‰, n=77)
also have similarity. These results show similar climate change in 3700 years ago and nowadays.
The life of *Tridacna* YX1 (11 years) is much shorter than the fossil *Tridacna* (which lived for at
least 40 years), thus, modern observation data were used to do the climatic comparison. After
translating $\delta^{18}O_{A5}$ into SST (Fig. 5), the reconstructed SST have an average maximum and
minimum of 30°C and 25.61 °C, respectively, with seasonal variation of 4.39 °C. Comparatively,
the r-monthly average range of modern observation is 29.33 to 23.99 °C (year from 1982 to 2017),
with seasonal variation of 5.34 °C. The warmer climate in the past indicates that the seasonality
variance is about 0.95 °C lower. Considering the seasonality discrepancy between $\delta^{18}O_{shell}$ and
$\delta^{18}O_{predicted}$, the $\delta^{18}O_{shell}$ has 19 % lower seasonal variation than $\delta^{18}O_{predicted}$. Therefore, the actual
seasonal variation of A5 (roughly 5.23 °C) is still below the present seasonality.
In addition, the discrepancy between mean $\delta^{18}O_{A5}$ and $\delta^{18}O_{YX1}$ is 0.19 ‰, the lower mean
$\delta^{18}O_{A5}$ is because of more r-months in lower values. This reveals a possible prolonged high
temperature period: Warm seasons may have been longer, while cold seasons are shorter. From the
r-monthly insolation comparison between 3700 years ago (3722 to 3623 BP) and recent decades
(1918 to 2017) (Fig. 4d), this coincides with the phenomenon that more insolation occurs from the
2$^{nd}$ to 5$^{th}$ r-month (warm seasons), yet less insolation occurs in the rest of the year. Due to the
more samples in *Tridacna* obtained in the warm seasons, the prolonged high temperature period
would be magnified (from the 2$^{nd}$ to 6$^{th}$ r-month) (Fig. 4c). Moreover, compared to the deviation
between the total average and r-monthly values, cold seasons have larger deviation and slope. This



illustrates a fast switching between cold and warm seasons in 3700 years ago. As $\delta^{18}O_{predicted}$ has
stronger seasonal variation than $\delta^{18}O_{shell}$, the slope should be sharper, means more significant
actual seasonal switching.

Overall, the climate in around 3700 years ago had slightly lower seasonality than present,

and the switching between cold to warm seasons was more serious.
*4.4 Climate variation comparison between 3700 years ago and present*

Global warming is considered to have triggered many disasters (Burgess et al., 2018;

Oppenheimer, 2008; Wang et al., 2015; R. Yu et al., 2018). Analogous studies on past warm
climate would allow us to better predict the future climate and extreme events if global warming
persists. Therefore, we compared modern instrumental observations (year from 1982 to 2017) in
the North Reef with the reconstructed SST anomalies of *Tridacna* A5. R-monthly resolution data
were first compared, which were obtained by subtracting the r-monthly SST with the mean value
of each r-monthly. In terms of long-term climatic variation, the SST anomalies are markedly
different between the 36-year modern instrument data and the 40-year reconstructed data (Fig. 6a).
The SST anomalies (3700 years ago) have sharper peaks and higher amplitude than in those of the
recent years, and the standard deviation in the past is much larger (0.68 °C) than the present
(0.42 °C), which suggest a more severe climate condition in the past. However, one has to be
aware of the different growth rates and equidistant sampling mode in *Tridacna*'s life when using
the r-monthly resolution. For example, *Tridacna* may have different annual growth rates, hence a
r-monthly value may not represent the corresponding actual r-monthly value under equidistant
sampling mode. In this respect, the r-annual SST anomalies are estimated (Fig. 6b). The SD of
modern observation is 0.30 °C, and the SD of reconstructed SST anomalies is 0.41 °C. This
illustrates that the ratios of the modern to the past in r-monthly resolution or r-annual resolution
are almost the same (0.65 and 0.73, respectively), thus the SD of r-monthly SST anomalies of
*Tridacna* is likely reliable. As a result, there was probably an enhanced climate variability 3700
years ago.
*4.5 ENSO activity recorded by Tridacna $\delta^{18}O$*

ENSO is the strongest signal in global interannual climate variation, and understanding its

mechanism is important to unravel the past climate change and forecast in the future one.





Interannual climate changes in the Xisha Islands were likely dominated by ENSO activity, and the
local SST anomalies may have reflected 76.47 % and 79.41 % on moderate El Ninõ and La Niña
events, respectively (Liu et al., 2016). Previous studies demonstrated that the marine biogenic
carbonate-based SST reconstructions in the northern South China Sea likely responded to ENSO
activity (Sun et al., 2005; Yan et al., 2017). Warm/cold SST anomalies were related to El Ninõ/La
Niña events. Coral is one of the earliest records for ENSO events (Peng et al., 2003; Sun et al.,
2005; Wei et al., 2007), yet there are still some technical limitations, such as those concerning the
calcite-affected data (McGregor and Gagan, 2003). Analyses on the *Tridacna* species were later
introduced to make up this imperfection, due to their denser shells, negligible diagenetic alteration,
and oxygen isotopic equilibrium with seawater. Recently, Yan et al. (2014) proved that *Tridacna*
species in the Xisha Islands could respond to ENSO activity, and then used fossil *Tridacna* $\delta^{18}O$ in
Dongdao Island (one of the islands in the Xisha Islands) to reconstruct ENSO variability around
2000 years ago (Yan et al., 2017).

To acquire more precise ENSO reconstructions, modern observation data were analyzed. The

SST of Ninõ 1 + 2 region was chosen due to the distinct seasonal variation as the same as the
study area, and the SST anomaly series were calculated by subtracting the r-monthly mean values
(seven points/yr). The spectral analyses were performed to test periodicity among all SST
anomalies (Fig. 7), which indicate spectral peak of three to seven years. According to the SST
series, the North Reef SST have a 3-month time lag behind the Ninõ 1 + 2 SST (Fig. 8a), and thus
we bring 3-month forward to eliminate the lag. To reconstruct the occurrence of ENSO-type in the
North Reef, 3-7 years bandpass filtering was performed on the SST anomalies, which yielded a
tendency of the North Reef ENSO activity mostly consistent with the Ninõ 1 + 2 SST anomalies
(Fig. 8c). We calculated a threshold value under 1σ SST anomalies for moderate El Ninõ/La Niña
events. A total of seven El Ninõ and ten La Niña events occurred in the past 36 years. In other
words, El Ninõ/La Niña events occurred successively in a 5.14-year frequency in the North Reef.

Spectral analysis revealed that $\delta^{18}O_{A5}$ anomalies also have a 3-7 years period (Fig. 7c). As

above discussed, the *Tridacna* $\delta^{18}O$ values are mainly dominated by SST in the Xisha Islands, and
1 ‰ $\delta^{18}O_{shell}$ is roughly equal to 4.41 °C of SST. We translate the $\delta^{18}O_{A5}$ anomalies into the North
Reef $SST_{A5}$ anomalies (Fig. 9b). After the 3-7 years bandpass filtering of the North Reef $SST_{A5}$



anomalies, six El Ninõ and five La Niña events were estimated to occur in 40 years with $1\sigma$ $SST_{A5}$
anomalies threshold (Fig. 9c), giving 6.67-year and 8-year frequency, respectively. The ENSO
frequency reduces when comparing with the modern observation data. The lower frequency
supported the ENSO reconstructions since 7 ka BP, which suggests a notable reduction of ENSO
between 5 ka BP and 3 ka BP (Liu et al., 2013; McGregor et al., 2013; Tudhope et al., 2001;
Emile-Geay et al., 2016). However, implications drawn from merely 40-year long *Tridacna* $\delta^{18}O$
record is likely inconclusive. Collection of more similar-age *Tridacna* is needed to acquire a more
continuous climate and ENSO activity record.
*4.6 Extreme winter El Ninõ records in fossil Tridacna $\delta^{18}O$ values*
Extreme El Ninõ brings about many climatic disasters, such as catastrophic flooding,
bushfire and drought, in recent decades (Ramírez and Briones, 2017; Staupe-Delgado et al., 2018;
Yu et al., 2018; Yu et al., 2019). With global warming persists, the question of whether high
temperatures are related to extreme El Ninõ events is still controversial. Therefore, records of
extreme El Ninõ events in the past warm periods are important. Here, the winter SST is used to
estimate the extreme El Ninõ events. Winters in the northern South China Sea are very dry, and
the SSS variation caused by rainfall is small. Thus, the SST determined from $\delta^{18}O$ should be close
to the actual value. The SST calculated by $\delta^{18}O_{YX1}$ reveal warmer winter in 1998, corresponding
to a stronger El Ninõ that year. Comparison between the reconstructed SST (calculated with
$\delta^{18}O_{A5}$) and modern observation data from the North Reef (Fig. 5), suggested that the average
winter SST in 3700 years ago was 25.62 °C. There are six distinctly high SST within the 40 years,
with the anomalies range from 0.73 to 2.00 °C. As for the SST of modern observation (year from
1982 to 2017), the average of winter SST is 23.99 °C, and three anomalously warm temperatures
vary from 0.6 to 1.38 °C. It seems that the extreme El Ninõ events occurred under higher
temperature and were more frequent in this past warm period. However, we still have low
confidence in answering this controversial question about the relationship between El Ninõ events
and warm climate, more *Tridacna* in the past warm period should be analyzed in future work.
Nevertheless, our results still put forward a high-resolution data that make a contribution to future
work on how El Ninõ performs in the warm period.



## 5 Conclusions

The $\delta^{18}O$ derived from *Tridacna* provide high-resolution data to unravel the climatic variability and ENSO activity. In the Xisha Islands of northern South China Sea, $\delta^{18}O_{shell}$ of modern *Tridacna gigas* can serve as a proxy of SST, while SSS has a minor effect on $\delta^{18}O_{shell}$. Thus, a $\delta^{18}O$-SST linear regression is established roughly: SST (°C) = 22.69 - 4.41 × $\delta^{18}O_{shell}$. Another *Tridacna squamosa* A5, which lived 3700 years ago, reveals 40 clearly dark/light couples consistent with $\delta^{18}O$ amplitude. Reconstructed SST implies a warmer climate in 3700 years ago, 0.84 °C higher than present. The seasonal variation slightly decreased and the switching among warm and cold seasons was faster. The combination of r-monthly-/r-annual-resolution reconstructed SST anomalies suggest an enhanced climatic variability during this past warm period. Besides, the frequency of ENSO activity reduced in 3700 years ago than that in recent 36-year modern observation. El Ninõ/La Niña events occurred alternatively in every 6.67-/8-year frequency in the past, compared to 5.14-year nowadays. The extreme winter El Ninõ has been recorded by fossil *Tridacna* under an increased and intense situation. Our results imply an unstable climate in 3700 years ago, although more data are still needed to support this hypothesis.

## Author Contributions

X. M. S., H. Y., Y. H. designed the research and experiments; H. Y. collected the samples; H. C., Y. H. performed stable isotope measurements. H. Y. and Y. H. did the data analyses. Y. H. wrote the manuscript, with the help of all co-authors.

## Competing interests

The authors declare that they have no conflict of interest.

## Data and materials availability

All data needed to evaluate the conclusions in the paper are presented in the paper. Additional data related to this paper may be requested from the authors. Correspondence and requests for materials should be addressed to X. M. S. (eessxm@mail.sysu.edu.cn) and H. Y. (yanhong@ieecas.cn).

## Acknowledgments

This work is supported by the projects from the National 13[th] Five Year Plan Project (DY135-R2-1-01, DY135-C1-1-06), National Nature Science Foundation of China (41877399,



41876038, 91128101, 41888101), State Key Laboratory for Mineral Deposits Research in Nanjing
University (No. 20–15–07), Chinese Academy of Sciences (QYZDB-SSW-DQC001), Qingdao
National Laboratory for Marine Science and Technology of China (QNLM2016ORP0202), and.
We are grateful to Dr. Yu Fu, Dr. Yang Lu, Dr. Jiaoyang Ruan, Chengcheng Liu, Tianjian Yang,
Jun Gu for their support in preparing the manuscript. Dr. Youfeng Ning, Hanying Li, Pengfei Duan,
Jingyao Zhao are thanked for their technical support in drilling and analyses in Xi'an Jiaotong
University.

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

**Figures**

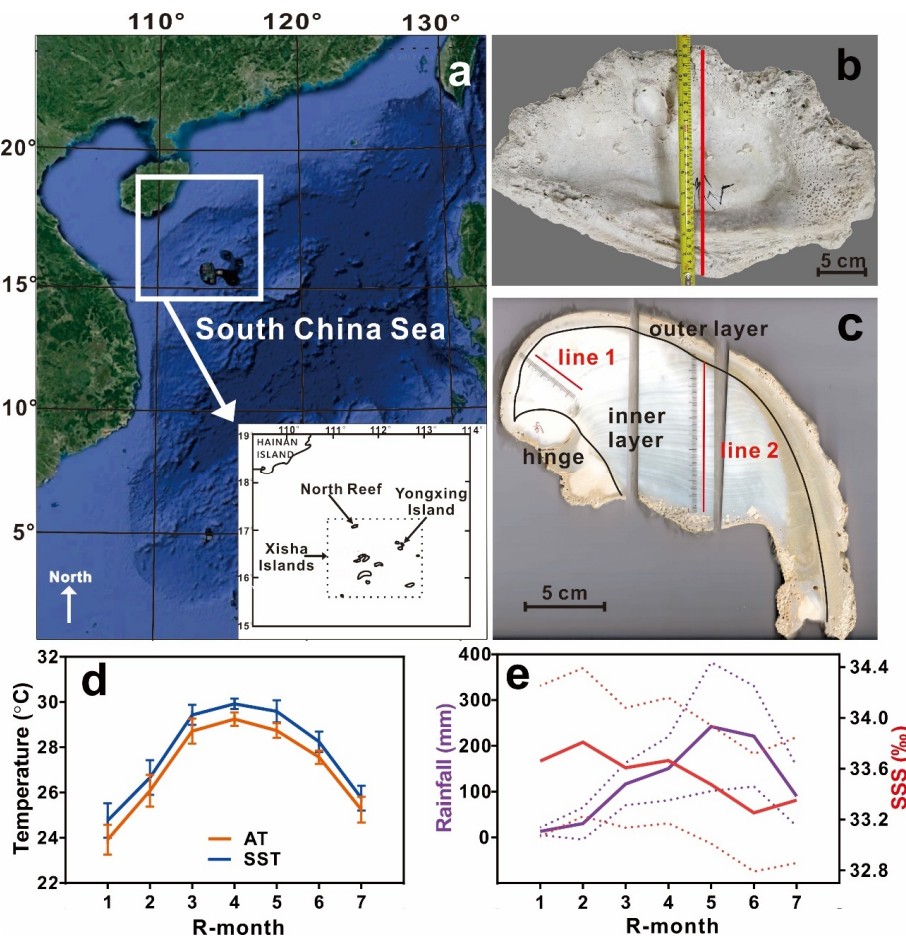


**Figure 1.** Maps of the South China Sea, with the location of the study area in the Xisha Islands (a).
Photo of *Tridacna* A5, and a slice was cut along the maximum growth axis (red line) from the
umbo to the ventral margin (b). Different parts can be seen clearly (hinge, inner layer, and outer
layer) (c), the red lines are the sampling lines for $\delta^{18}O$ analysis. Meteorological observations in the
Xisha Islands from 1994 to 2005: R-monthly average air temperature (AT) and sea surface
temperature (SST) (d); R-monthly average rainfall and sea surface salinity (SSS) with standard
deviation (1σ) (e).

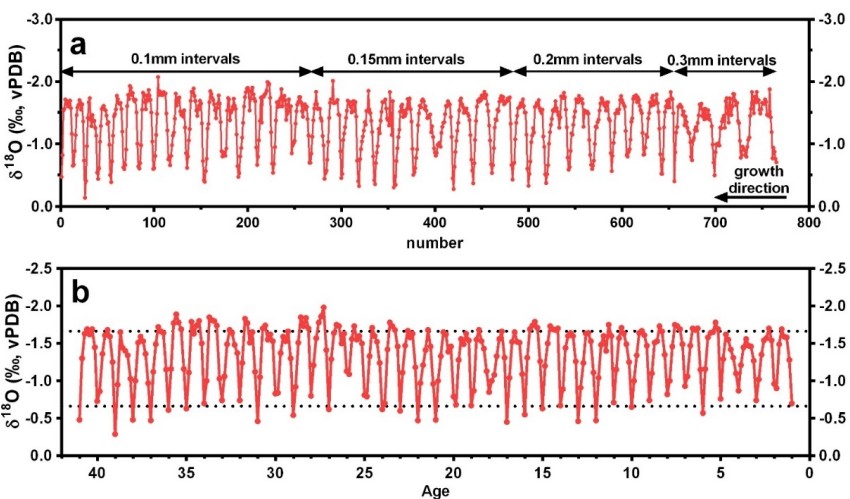

**Figure 2.** The δ[18]O profiles of A5 (a). The δ[18]O$_{A5}$ series with chronology time-scale after
rehandling data, and the dotted lines indicate the average of annual maximum and minimum (b).

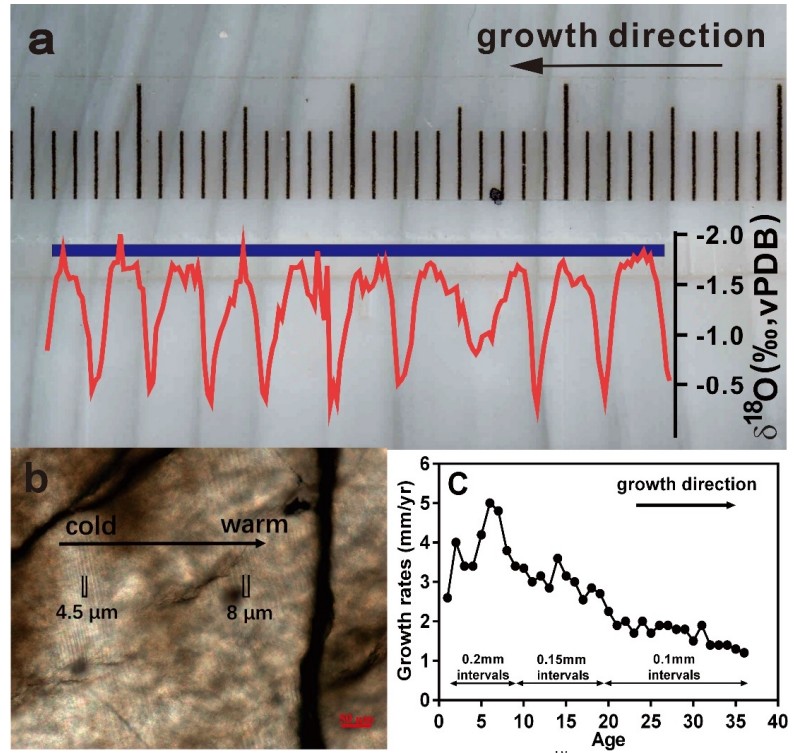

**Figure 3.** Amplitude of dark/light couples, consistent with $\delta^{18}O_{A5}$ profiles. Dark and light
increments correspond to high $\delta^{18}O$ (cold seasons) and low $\delta^{18}O$ (warm seasons). Blue line
represents the sampling line (a). Under the microscope, daily increments grow slower in cold
seasons, but faster in warm seasons (b). Growth rates (line 2) in fossil *Tridacna* A5 (c).

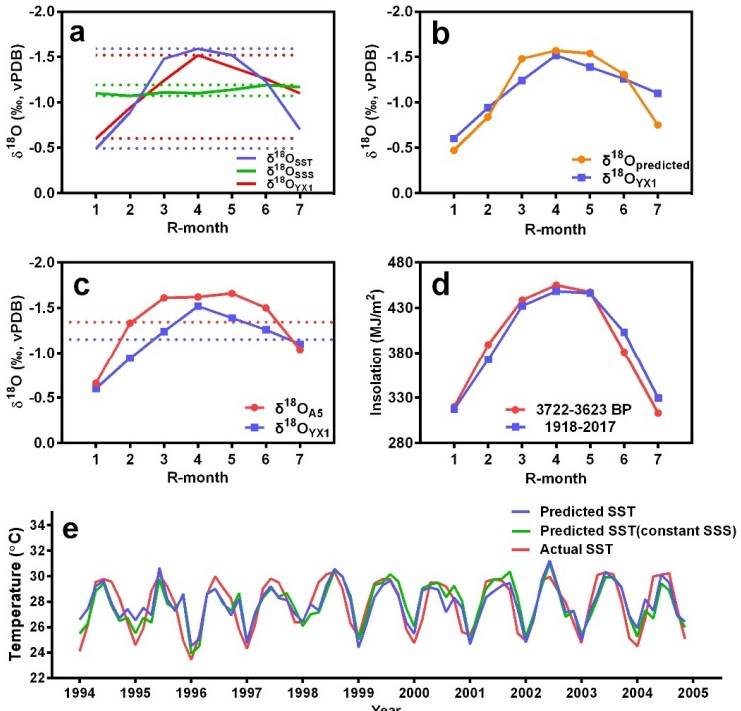


**Figure 4.** Predicted r-monthly $\delta^{18}O$ profiles under constant SSS (blue line) and constant SST

(green line) conditions, and $\delta^{18}O$ of YX1 (red line). Dotted lines represent the maximum and

minimum of the r-monthly $\delta^{18}O$ profiles (a). R-Monthly average $\delta^{18}O_{YX1}$ and $\delta^{18}O_{predicted}$ (b).

R-monthly average $\delta^{18}O_{YX1}$ and $\delta^{18}O_{A5}$, and the dotted lines represent mean values (c). Different

insolation in 3700 years ago and in the recent 100 years (d). Different SST profiles: predicted SST

with varied SSS (blue line), constant mean SSS (green line), and actual SST (red line) (e).

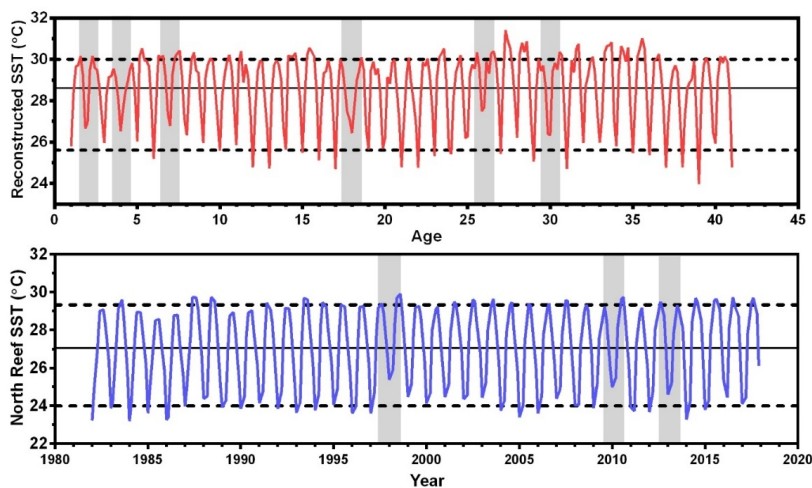

**Figure 5.** Reconstructed SST around 3700 years ago (red), compared with the North Reef SST
from 1982 to 2017 (blue). Dotted lines represent the average maximum and minimum SST. Gray
field represents the extreme winter El Ninō events.

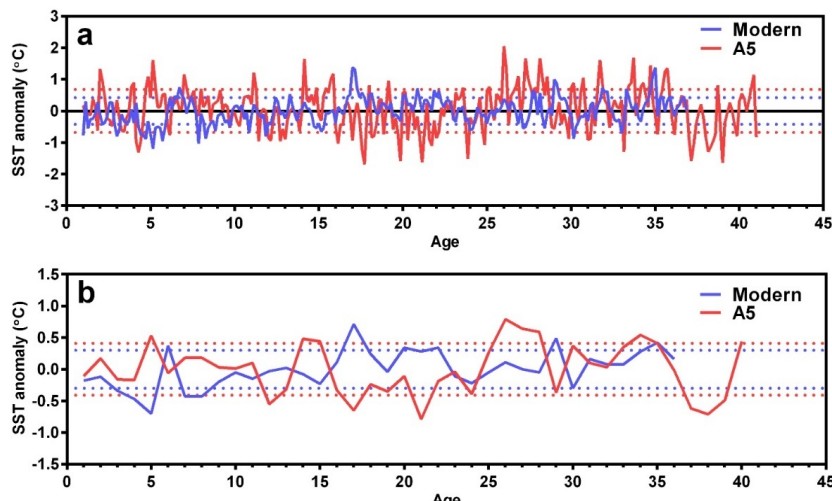

**Figure 6.** SST anomalies of modern instrument data and reconstructed SST anomalies of *Tridacna*
A5 under r-monthly (a) and r-annual (b) resolution. Dotted lines represent one standard deviation
(1σ) of SST anomalies.

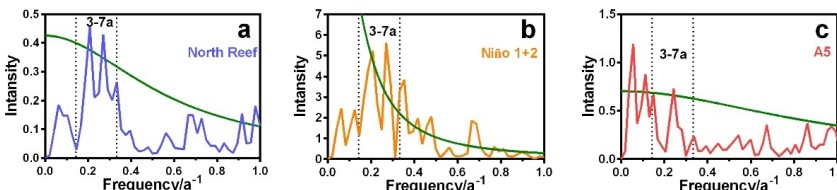

**Figure 7.** Spectral analysis of the North Reef SST anomalies (a), Ninõ 1 + 2 SST anomalies (b),
and reconstructed SST anomalies according to $\delta^{18}O_{A5}$ (c). Green lines indicate significant lines at
90 % confidence level, and the area between two dotted lines represents the frequency from 3 to 7
years.

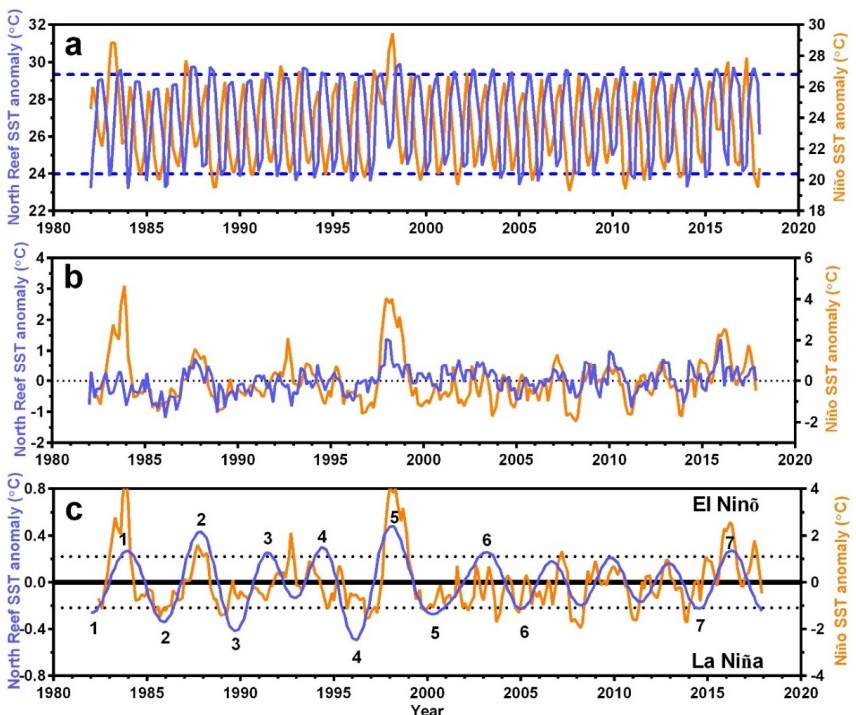


**Figure 8.** Relationship between ENSO activity and the North Reef SST: The North Reef SST
(blue line) compared with Ninõ 1 + 2 SST (yellow line), a clear time lag exists (a). SST anomalies
of two areas, and the lag is removed by forwarding the North Reef SST anomalies for 3 r-months
(b). The North Reef SST anomalies performed with 3-7 years bandpass filter, consistent with Ninõ
1 + 2 SST anomalies, and the dash lines show the calculated threshold limits ($1\sigma$) for El Ninõ and
La Niña events in the North Reef (c). El Ninõ and La Niña events are represented by positive and
negative SST anomalies values, respectively.

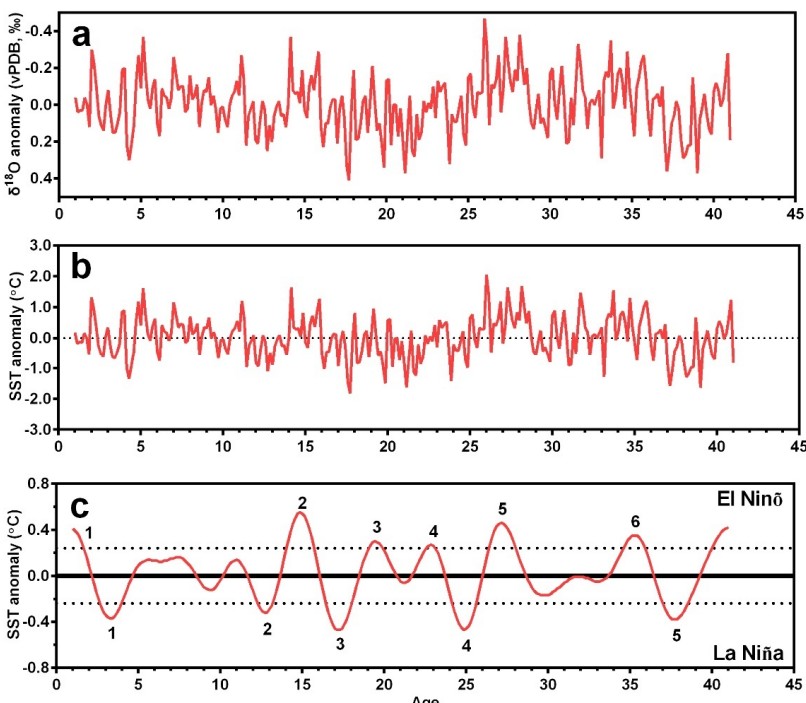

**Figure 9.** ENSO activity reconstructed by fossil *Tridacna* 3700 years ago: $\delta^{18}O$ anomalies of fossil *Tridacna* A5 (a). The North Reef SST anomalies calculated by $\delta^{18}O$ anomalies (b), based on modern *Tridacna* $\delta^{18}O$-SST equation (1 ‰ $\delta^{18}O_{shell}$ ≈ 4.41°C SST). ENSO activity according to the North Reef SST anomalies after 3-7 years bandpass filtering, and the dash lines show the calculated threshold limits (1σ) for El Ninõ and La Niña events (c).