# Peer review of "Evidence from giant clam δ18O of intense El Ninõ-Southern"

_Climate of the Past, 2019_

## Referee Comment (RC1) · Anonymous Referee #1 · 8 Aug 2019

The article presents a new oxygen stable isotope 40-year long record of a 3700 yr BP fossil giant clam Tridacna from the South China Sea. The fossil record is compared to a modern tridacna shell and instrumental data. The authors show clearly that the shells faithfully record SST variations with a nearly monthly resolution. The sclerochronological work is precise and performed with caution. Great attention was given to the effect of sampling resolution. All record were resampled at the same resolution for better comparison of SST ranges and variability. The fossil tridacna shell recorded EN SO variability as shown by the spectral analysis and the 3-7 filtered signal. ENSO signal showed a slightly lower frequency and stronger events at 3700 cal BP compared to the modern reference period. As the author acknowledge, the studied period is too short to draw conclusion on ENSO variability but the study provides high quality new

paleoclimate data. Such quantitative seasonally resolved datasets are necessary to achieve a more detailed understanding of the relationship between long-term background changes and seasonal to interannual climate variability.

I consider therefore that this is a valuable contribution that needs to be published with minor corrections. The text requires some work with the English. It is generally OK to be read and understood, except for a few sentences that I mention hereafter, but it contains numerous grammatical, syntax and vocabulary errors that need to be fixed. I did not note all the English errors because that is beyond a reviewer's work. In any case, languages issues should not prevent this paper from being published. I hope the journal can assist the authors with language edition. Besides this, the introduction and discussion should include a more complete bibliography of paleo-ENSO reconstruction. Key papers such as Koutavas et al. Paleoceanography (2012), Cobb et al. Science (2013), Carré et al., Science (2014) are neither cited nor discussed. A substantial part of the results and discussion is dedicated to changes in the SST seasonality. A new figure showing average seasonal cycles (mean and s.d.) from the fossil, modern, and instrumental record would summarize and clarify greatly the result.

Detail comments:

L59-60: "ontogenic reduction": do you refer to the decreasing growth rate with ontogeny?

L65: "uncertainties": did you mean "unclear"?

L76: "involved in Holocene Megathermal period": did you mean "part of the Holocene climatic optimum"?

L86: "trigger" should be "source of"

L91: Clement et al., 1999 is a modeling study, not a reconstruction.

L93-98: incomplete bibliography.

L123-125: "Due to . . . actual month". This sentence needs to be rewritten. I understood that the records were resampled at 7 data points per year to have comparable time resolution across the records. This number was chosen because it corresponds to the lowest resolution achieved in the fossil record. The verb "rehandle" is used throughout the manuscript but I think "resample" would be more appropriate and clearer. What technique was used for the resampling? Linear interpolation?

L144-146: some clarification is needed about the radiocarbon date calibration. What DR value was used? "Conventional" cannot refer to the calibrated date. The calibrated date should not have a +/-28 year uncertainty. Calibration yields a 1sigma or 2 sigma confidence interval and a median date.

L163-167: this part is unclear. Are you comparing values of the internal standards obtained during the analyses of both shells? Is it the same standard material?

L170-171: "which contained. . .life span". This is unclear

L192: "daily increments are obvious". They are not to me on the figure. Clarify

L226: "perfect match, r=0.81". perfect sounds too strong. Why is d18O(XY1) better correlate to d18O(SST) (r=0.91) than to d18O(predicted) (r=0.81) if this latter includes both SST and SSS and should therefore be more realistic?

L214 – L240: these paragraphs could be shorter and clearer if the information was better organized and presented.

L244: "variance" . Do tou refer to the seasonal range?

L244: 0.19% check this number.

L277: "ndcates" do you mean "associated with"?

L269-280: The total range of the signal includes not only seasonality but also interannual to decadal variability. To evaluate the change in the seasonal range, it would be more appropriate to estimate and compare the mean seasonal ranges.

L288: "Moreover...slope" this is unclear

L290-292: a figure of mean seasonal cycles would be useful

L296-299: these short introductions about global warming are not necessary

L320: unclear

L333: unclear

---

## Referee Comment (RC2) · Anonymous Referee #2 · 19 Nov 2019

General comments:

Hu et al. present a new oxygen stable isotope record of a fossil giant clam from the South China Sea, which reveals new high resolution insights into the ENSO activity dated back 3700 yr BP and fine-tuned using a modern Tridacna for comparison. As this study fits well into the journal's scope I rate this manuscript to be of high interest to the audience of Climate of the Past and encourage publication after minor revision. As the study was carried out on only one specimen it has a "case study-like" read, however, the authors convince me that their application bears high potential for a potential larger-scale study with more specimens. The manuscript is well structured and outlined. The methodological part appears sound, which is apparent when e.g. sampling resolutions are discussed. I feel the introduction could benefit from discussing

and citing more sclerochronological papers discussing oxygen stable isotope records from bivalves (they don't have to relate to the sampling site) and I would strongly argue that a recent paper demonstrating shell architecture of Tridacna ought to be mentioned and cited (Agbaje et al.2017). Further, I have some comments to the title (see below) and there are a few other (mostly language) issues that I feel need fixing before moving forwards and I provide a list of more detailed comments below to address these. I enjoyed reading this study and hope the authors will find my suggestions helpful and encouraging!

Specific comments:

L1-2: I believe the use of "ENSO" in the title is not wise. Titles should be fully understandable to a broad audience and community-specific abbreviations should be avoided. I'd urge the authors to type out "ENSO" or phrase this differently. Also it may be good to use "Giant Clam" instead of "Tridacna" in the title.

L22: "are the largest marine bivalves" and "carbonaceous shell" and "can be used for high-resolution paleoclimate reconstructions". L47: delete "of".

L48: "physicochemical" is weird in this context – do you want to record environmental signatures encoded within the biocarbonate or do you want to look at physiological variations that may or may not be influences by external factors?

L49: "on past climate dynamics" delete "the".

L50-51: I recommend also citing the most recent work on the crossed-lamellar shell architecture of Tridacna see reference: Agbaje, O. B. A., R. Wirth, L. F. G. Morales, K. Shirai, M. Kosnik, T. Watanabe, and D. E. Jacob. "Architecture of crossed-lamellar bivalve shells: the southern giant clam (Tridacna derasa, Röding, 1798)." Royal Society open science 4, no. 9 (2017): 170622.

L54: I doubt that Tridacna lives up to "few centuries" where is the evidence (reference)? This may have been mixed up with Arctica shells or other long-lived bivalves but these

are very different from Tridacna!

L57: "precipitate" is really a wrong term when talking about shells as it is closely associated with classical crystallisation pathways (i.e. "inorganic" systems). However, we know for more than over a decade now that shells form by non-classical crystallisation pathways via precursor phases (amorphous calcium carbonate and/or vaterite). I am not saying you need to venture into the area of shell biomineralization here but I would strongly argue to find a better word for this text passage. Maybe replacing "precipitate their shells" with simply "grow".

L59-60: What do you mean with "ontogenetic reduction"?

L80: "occurring nowadays", however, I think you should try and find a more appropriate word than "nowadays" as this sounds perhaps too casual and please replace throughout manuscript.

L83-84: Better: "High-resolution isotopic geochemical data from Tridacna may provide detailed insight into climatic variations of this period."

L117: "give distinct seasonal SST to the Tridacna from the coral reefs" reads clumsy, perhaps change to "provide distinct seasonal SST for Tridacna populating the coral reefs of the Xisha Islands".

L123-125: I don't understand "rehandling" do you mean "re-sampling"? I agree with referee 1 that this sentence needs to be rewritten for more clarity. Please change throughout the manuscript.

L130-131: Perhaps better: "It is excluded that river runoff effects SSS as the Xisha Islands are at a XXX km distance to the continental mainland." Please quantify roughly to provide evidence.

L138-143: I recommend providing a sentence regarding the crossed-lamellar shell architecture of Tridacna see above mentioned reference Agbaje et al. (2017).

L144: when you mention "14C AMS" for the first time I recommend providing the full method name in brackets (replace "14C AMS" with "14C AMS (Accelerator Mass Spectrometry)") for readers that lack this methodological background.

L145: I don't understand the meaning of "conventional" in this sentence – maybe not the right phrase? What is the uncertainty? First or second standard deviation or something else?

L154: "from adult to childhood" is not the right phrase how about "in a transect from adult to ontogenetically younger shell"?

L185: "40 dark/light couples (each representing one year)" please explain how darklight line couples relate to time/tide schedules/seasonality. How much time/which tide pattern does one ark-light line couple stand for?

L192: Increments are not obvious to me from the image. Especially Fig. 3b is not clear what one should see, perhaps choose a different image with better resolution.

L192-193: "In general, Tridacna A5 grew faster in warm seasons and slower in cold seasons (Fig. 3b)." Where is your evidence for this assumption? I feel you need to back this up as this varies between species and you need to demonstrate to the reader that it is the case for Tridacna. Also more seasonal information may be needed to achieve this. How long are summers how long are winters? For example: if a reader believes summer and winter are similar in length one could misinterpret short low $\delta18O$ periods may have just been formed quicker (and have thus higher not lower growth rates!). This all needs more explanation and demonstration and is important as you build upon this later in the discussion. Perhaps see other papers I suggest any study by Carré et al as they are very educative in this respect.

L196-197: I don't understand this sentence.

L201: Perhaps not everything about Tridacna but $\delta18O$?

L259: "lived 3700 years ago" delete "in".

L286-287: Better: "Due to a higher sampling density in Tridacna. . .".

L288: "magnified" is the wrong word here.

L292: "switching" wrong word, replace throughout manuscript.

L293-294: This sentence contradicts itself and needs rewording for clarity.

L296-299: reads more like an introduction section and is not relevant here (suggest to delete).

L303: "instrumentation data" is odd.

L326: "calcite-affected" sounds also a bit odd to me maybe you can find a better term. Why is calcite "bad" in this sense? Why is it a limitation?

L326-328: Better perhaps: "Analyses of Tridacna species are performed to overcome this limitation by taking advantage of their denser shells, negligible diagenetic alteration, and oxygen isotopic equilibrium with seawater."

L338: unclear.

Figure 1: It looks like your 5 cm scale bar is too large for the scale in the figure (measuring tape, here 5 cm look smaller). There are some grammar issues in the figure caption.

L633: "amplitude" may not be the right word here.

L635-636: "under the microscope, daily increments grow slower in cold seasons, but faster in warm seasons" – this is not visi8ble from microscope images alone! This needs more explanation! Also, image is not really easy to understand (what should be seen? It's all very blurry).

---

## Author Comment (AC1) · 10 Dec 2019

*Anonymous Reviewer #1:*

*We would like to thank this reviewer for her/his comments on our manuscript. Please find our detailed answers to each comment below. The reviewer comments are in normal black script, our answers are in blue italics and the revised texts are in blue normal script.*

General Comments:

The article presents a new oxygen stable isotope 40-year long record of a 3700 yr BP fossil giant clam Tridacna from the South China Sea. The fossil record is compared to a modern tridacna shell and instrumental data. The authors show clearly that the shells faithfully record SST variations with a nearly monthly resolution. The sclerochronological work is precise and performed with caution. Great attention was given to the effect of sampling resolution. All records were resampled at the same resolution for better comparison of SST ranges and variability. The fossil tridacna shell recorded ENSO variability as shown by the spectral analysis and the 3-7 filtered signal. ENSO signal showed a slightly lower frequency and stronger events at 3700 cal BP compared to the modern reference period. As the author acknowledge, the studied period is too short to draw conclusion on ENSO variability but the study provides high quality new paleoclimate data. Such quantitative seasonally resolved datasets are necessary to achieve a more detailed understanding of the relationship between long-term background changes and seasonal to interannual climate variability.

*We thank the reviewer for her/his positive evaluation of our manuscript.*

I consider therefore that this is a valuable contribution that needs to be published with minor corrections. The text requires some work with the English. It is generally OK to be read and understood, except for a few sentences that I mention hereafter, but it contains numerous grammatical, syntax and vocabulary errors that need to be fixed. I did not note all the English errors because that is beyond a reviewer's work. In any case, languages issues should not prevent this paper from being published. I hope the journal can assist the authors with language edition.

*Thanks for the suggestion, we have checked the errors in our best to improve our manuscript.*

Besides this, the introduction and discussion should include a more complete

bibliography of paleo-ENSO reconstruction. Key papers such as Koutavas et al. Paleoceanography (2012), Cobb et al. Science (2013), Carré et al., Science (2014) are neither cited nor discussed.

*Thank you very much for the classical references. We have added them to our induction and clarified in brief.*

A substantial part of the results and discussion is dedicated to changes in the SST seasonality. A new figure showing average seasonal cycles (mean and s.d.) from the fossil, modern, and instrumental record would summarize and clarify greatly the result.

*We agree with the reviewer, a figure (Fig. 4e) have added with average seasonal cycles from the data and clarified the result in our manuscript. We only use the data of fossil Tridacna and the modern instrumental record in the figure. This is because they are both in North Reef. Modern Tridacna is located in Yongxing Island (the southern of North Reef), therefore, the seasonality may have deviation which is inappropriate to compare with data in North Reef. To avoid the deviation, we think the average seasonal cycles (minimum, maximum, s.d.) of fossil Tridacna and North Reef instrumental record is better to summarize and clarify the result.*

[Figure]

Fig. 4

Detail comments:

L59-60: "ontogenic reduction": do you refer to the decreasing growth rate with

ontogeny?

*Yes, as K. Welsh (2011) indicated in his article, the ontogenic reduction in growth of Tridacna gigas does not reduce the reliability with which temperature and $\delta^{18}O_w$ variability can be reconstructed. Climate reconstruction in $\delta^{18}O_{shell}$ does not have an incongruity with temperature and $\delta^{18}O_w$ which might be an obviously declined or increased tendency. We rephrased this sentence to make this clearer:*
and the reliability in reconstruction between temperature and $\delta^{18}O_w$ variability would not be reduced by the ontogenic reduction in growth of the *Tridacna* $\delta^{18}O$.

L65: "uncertainties": did you mean "unclear"?

*Yes, we have corrected this.*

L76: "involved in Holocene Megathermal period", did you mean "part of the Holocene climatic optimum"?

*Yes, we have changed the expression to make it clear in the manuscript.*

L86: "trigger" should be "source of"

*Thank you for your advice. We have corrected this.*

L91: Clement et al., 1999 is a modeling study, not a reconstruction.

*Thank you. We have corrected it in the text.*

L93-98: incomplete bibliography.

*Thanks for the suggestion and classical references commends. We have added them to the text and clarified as followed:*
Furthermore, the El Ninõ-Southern Oscillation (ENSO) is widely accepted to be the main source of interannual climatic variability in the Pacific Ocean. Previous studies suggested that the impacts of ENSO activity would not be limited to the tropical area, but also on the global atmospheric circulation through heating-up of the tropical atmosphere (Cane, 2005). Thus, the reconstruction of ENSO is very important for understanding its dynamics and predicting future change. Many early published ENSO behaviors were constructed with low-resolution proxy data using deposition events (Rodbell et al., 1999; Koutavas and Joanides, 2012), ice cores (Thompson et al., 1995;1998), to reveal the ENSO variance in thousands of years. However, the periodicity of ENSO was short compared to those low-resolution data, it's difficult to demonstrate the strength and variability of ENSO activity. Recent studies focus on seasonal or monthly data to examine the precise variation in ENSO activity (Arias-Ruiz et al., 2017; Ayling et al., 2015; McGregor et al., 2013; Welsh et al., 2011; Yan et al.,

2017), but those fragmental data cannot fully understand the Holocene ENSO dynamics. Therefore, fragments at different times according to different high-resolution samples are needed and can provide an integrated framework for examining ENSO theory and models in Holocene. Besides, studies on the mid to late Holocene ENSO evolution yielded controversial findings: Coral records showed a reduced ENSO variability around the late Holocene (McGregor et al., 2013; Cobb et al., 2013; Woodroffe et al., 2003). Other carbonate species like fossil mollusk shells suggested that ENSO variance was severely damped ~4000 years ago (Carré et al., 2014). Yet some other studies indicated a strengthening ENSO activity at 4 to 3 ka BP (Tudhope et al., 2001; Duprey et al., 2014; Yang et al., 2019). Thus, this further points to the importance of high-resolution isotopic geochemical data such as *Tridacna* in unraveling the dynamics of ENSO.

L123-125: "Due to ⋯⋯ actual month". This sentence needs to be rewritten. I understood

that the records were resampled at 7 data points per year to have comparable time resolution across the records. This number was chosen because it corresponds to the lowest resolution achieved in the fossil record. The verb "rehandle" is used throughout the manuscript but I think "resample" would be more appropriate and clearer. What technique was used for the resampling? Linear interpolation?

*Thanks for the suggestion. We agree with you that change the verb "rehandle" into "resample" will be better. The technique we used for the resampling is a cubic spline model in AnalySeries 2.0.8. This method was first applied by Schöne and Fiebig (2009), who used bivalve shells (Arctica islandica) to reconstruct climate. They suggested that 7 points per month would elapse during the core growing season of the shell (i.e., time interval of fastest shell growth covering the seasonal extremes). And only the annual sample number for which equal to or more than seven existed could be used. Therefore, we used 7 points per month.*

L144-146: some clarification is needed about the radiocarbon date calibration. What DR value was used? "Conventional" cannot refer to the calibrated date. The calibrated date should not have a +/-28 year uncertainty. Calibration yields a 1sigma or 2 sigma confidence interval and a median date.

*From the modern Tridacna samples we collected in this area, the dating results showed no obvious "reservoir effect" (Liu et al., 2019). Tridacna might exchange its carbon with the atmosphere through photosynthesis. Therefore, we used the atmospheric $^{14}C$ yield model to calibrated. We have clarified the details about the radiocarbon date calibration as followed:*
The radiocarbon age determination was performed at Institute of Earth Environment of Chinese Academy of Sciences. The $^{14}$C Accelerator Mass Spectrometry data revealed the fossil *Tridacna gigas* age was $3437 \pm 28$ yr BP. Due to no obvious "reservoir effect" in dating results of modern *Tridacna* shells, the atmospheric $^{14}$C yield model was used

to calibration. The calibrated date (2σ) was range from 1783 to 1663 cal BC, with the median date is 1741 cal BC by using the IntCal13 of Radiocarbon Calibration Program CALIB 7.10.

L163-167: this part is unclear. Are you comparing values of the internal standards obtained during the analyses of both shells? Is it the same standard material?

*The two standard materials are the same material with a different name, we have corrected both of them into "GBW04405" in the text to avoid confusion. This ordovician carbonate is from Zhoukoudian Country, Beijing, China. The certified value and standard deviation had been obtained depend on the comparing with international standard NBS-19, and the result is in ‰ relative to VPDB.*

L170-171: "which contained······life span". This is unclear

*What we want to express is that the 100 years contain the probable life span of Tridacna when considering the 2 sigma confidence interval of corrected age. We have clarified it clear in the manuscript.*

L192: "daily increments are obvious". They are not to me on the figure. Clarify

*We have retreated the picture and clarified the details of the daily increments in the text.*

[Figure]

Figure 3. (a) Dark/light lines consistent with $\delta^{18}O_{A5}$ profiles. Dark and light lines correspond to high $\delta^{18}O$ (cold seasons) and low $\delta^{18}O$ (warm seasons), respectively. The distance between the dash lines represents a year that *Tridacna* grew. Blue line represents the sampling line. (b) Under the microscope, daily increments (a dark coupled with a light increment) grow slower when seasons are cold, but faster when temperature rises up. (c) Growth rates (line 2 in Fig. 1c) in fossil *Tridacna* A5.

L226: "perfect match, r=0.81". perfect sounds too strong. Why is $\delta^{18}O_{YX1}$ better correlate to $\delta^{18}O_{SST}$ (r=0.91) than to $\delta^{18}O_{predicted}$ (r=0.81) if this latter includes both SST and SSS and should therefore be more realistic?

*Thank you for your advice, we have rewrote this phrase.*
*$\delta^{18}O_{SST}$ were calculated in actual varied SST and constant SSS, $\delta^{18}O_{predicted}$ were calculated in both actual varied SST and SSS. Theoretically speaking, $\delta^{18}O_{YX1}$ should have a better correlation to $\delta^{18}O_{predicted}$. But our results are not. These might due to the significant control on SST to $\delta^{18}O_{shell}$. $\delta^{18}O_{SSS}$ are nearly negative correlated with $\delta^{18}O_{SST}$. This might reduce the correlation when using both actual varied SST and SSS to calculate.*

L214 – L240: these paragraphs could be shorter and clearer if the information was better organized and presented.

*We have rewrote these paragraphs into:*
*The $\delta^{18}O_{shell}$ reflect a combination of SST and SSS variation. In order to quantify more precisely what extent those factors influence on $\delta^{18}O_{shell}$ composition, two $\delta^{18}O$ profiles are calculated: $\delta^{18}O_{SST}$ (constant SSS but varying SST) and $\delta^{18}O_{SSS}$ (constant SST but varying SSS) (Fig. 4a). R-monthly mean values are used to minimize the influence of extreme events. The results show $\delta^{18}O_{YX1}$, $\delta^{18}O_{SST}$, and $\delta^{18}O_{SSS}$ profiles are range from -0.57 to -1.52 ‰, -0.48 to -1.58 ‰, -1.07 to -1.19 ‰, respectively. It is obviously that $\delta^{18}O_{YX1}$ and $\delta^{18}O_{SST}$ have same trend and high correlation (r = 0.91, n = 7; r = 0.78, n = 77), but the variation range in $\delta^{18}O_{SSS}$ is only 14 % of $\delta^{18}O_{YX1}$. Therefore, this indicates $\delta^{18}O_{shell}$ in the Xisha Islands correspond predominantly to the seasonal SST variation. Besides, the calculated $\delta^{18}O_{predicted}$ (by using both local actual SST and SSS) were used to compare with $\delta^{18}O_{YX1}$ (Table S1). The $\delta^{18}O_{YX1}$ and $\delta^{18}O_{predicted}$ profiles have nearly the same mean value (1.15 ‰ and 1.14 ‰, respectively). And their positive correlation (r = 0.81, n = 77) indicates the local Tridacna precipitates its shell in oxygen isotopic equilibrium.*
*Moreover, the comparison of predicted SST (under constant SSS and actual SSS with $\delta^{18}O_{YX1}$) further confirms that the SSS variation have no significant affection in local reconstructed SST (Fig. 4f). Two predicted SST values have high similarity (r = 0.93), and they are well correlated with the actual SST ($r_{vary}$ = 0.79, $r_{constant}$ = 0.78). Thus, we can use $\delta^{18}O_{shell}$ to roughly estimate the seasonal local SST variation, and establish a new SST-$\delta^{18}O_{shell}$ linear regression: SST (°C) = 22.69 - 4.41 × $\delta^{18}O_{shell}$ (or $\delta^{18}O_{shell}$ (‰) = -0.136 × SST + 2.634). A 1 ‰ change of $\delta^{18}O_{shell}$ is roughly equal to 4.41°C of SST. Yu (2005) summarized many published $\delta^{18}O$-SST slopes for the other marine carbonate species, Porites lutea coral, and suggested that the slopes could range from -0.134 to -0.189. Corals from Hainan Island revealed a good $\delta^{18}O$ vs. SST correlation with a linear regression slope of -0.137 (Su et al., 2006), very similar to our result (-0.136). Consequently, it is reliable to use the new linear regression for reconstructing the past SST with the fossil $\delta^{18}O_{shell}$.*

L244: "variance". Do you refer to the seasonal range?

*We have rephrased this sentence in the text to clarify:*
*The difference in seasonality between $\delta^{18}O_{YX1}$ and $\delta^{18}O_{predicted}$ is 0.18 ‰, which is accounts for 19 % of $\delta^{18}O_{YX1}$.*

L244: 0.19% check this number.

*Thank you. The number should be 19 %. We have corrected it in the text.*

L277: "indicates" do you mean "associated with"?

*Yes, we have corrected this.*

L269-280: The total range of the signal includes not only seasonality but also interannual to decadal variability. To evaluate the change in the seasonal range, it would be more appropriate to estimate and compare the mean seasonal ranges.

*Thank you for your advice, we have added this figure in Fig. 4e (see above).*

L288: "Moreover ····· slope" this is unclear

*We have rewrote the sentence:*
Moreover, comparing from each r-monthly value to average value, cold seasons have a larger deviation with greater slope.

L290-292: a figure of mean seasonal cycles would be useful

*We have added this figure in Fig. 4e.*

L296-299: these short introductions about global warming are not necessary

*Thanks for your suggestion, we have removed these sentences.*

L320: unclear

*We have rewrote this sentence to:*
The local accumulated positive percentage of monthly SST anomalies threshold could respond to 76.47 % El Ninõ and 79.41 % La Niña events in Ninõ 3.4 region (Liu et al., 2016).

L333: unclear

*We have rewrote those sentences to:*
To acquire more precise ENSO reconstructions, modern observation data were analyzed and compared with the SST in Ninõ 1 + 2 region. The SST anomaly series were calculated by subtracting the r-monthly mean values.

*Reference*
Schöne, B. R. and Fiebig, J.: Seasonality in the North Sea during the Allerød and Late Medieval Climate Optimum using bivalve sclerochronology, Int. J. Earth Sci., 98(1), 83–98, doi:10.1007/s00531-008-0363-7, 2009.
Welsh, K., Elliot, M., Tudhope, A., Ayling, B. and Chappell, J.: Giant bivalves (*Tridacna gigas*) as recorders of ENSO variability, Earth Planet. Sci. Lett., 307(3–

4), 266–270, doi:10.1016/j.epsl.2011.05.032, 2011.

Liu, C., Yan, H., Fei, H., Ma, X., Zhang, W. and Shi, G.: Journal of Asian Earth Sciences Temperature seasonality and ENSO variability in the northern South China Sea during the Medieval Climate Anomaly interval derived from the Sr / Ca ratios of *Tridacna* shell, J. Asian Earth Sci., 180(June), 1-9, doi:10.1016/j.jseaes.2019.103880, 2019.

---

## Author Comment (AC2) · 10 Dec 2019

*Anonymous Reviewer #2:*
*We would like to thank this reviewer for her/his careful reading on our manuscript. Please find our detailed answers to each comment below. The reviewer comments are in normal black script, our answers are in blue italics and the revised texts are in blue normal script.*

General Comments:

Hu et al. present a new oxygen stable isotope record of a fossil giant clam from the South China Sea, which reveals new high resolution insights into the ENSO activity dated back 3700 yr BP and fine-tuned using a modern Tridacna for comparison. As this study fits well into the journal's scope I rate this manuscript to be of high interest to the audience of Climate of the Past and encourage publication after minor revision. As the study was carried out on only one specimen it has a "case study-like" read, however, the authors convince me that their application bears high potential for a potential larger-scale study with more specimens. The manuscript is well structured and outlined. The methodological part appears sound, which is apparent when e.g. sampling resolutions are discussed. I feel the introduction could benefit from discussing and citing more sclerochronological papers discussing oxygen stable isotope records from bivalves (they don't have to relate to the sampling site) and I would strongly argue that a recent paper demonstrating shell architecture of Tridacna ought to be mentioned and cited (Agbaje et al.2017). Further, I have some comments to the title (see below) and there are a few other (mostly language) issues that I feel need fixing before moving forwards and I provide a list of more detailed comments below to address these. I enjoyed reading this study and hope the authors will find my suggestions helpful and encouraging!

*We thank the reviewer for his/her positive evaluation of our manuscript, and the detailed comments he/she suggested are really helpful. We have checked those errors to improve our manuscript and answered the questions in detail below.*

Specific comments:

L1-2: I believe the use of "ENSO" in the title is not wise. Titles should be fully understandable to a broad audience and community-specific abbreviations should be avoided. I'd urge the authors to type out "ENSO" or phrase this differently. Also it may be good to use "Giant Clam" instead of "Tridacna" in the title.

*Thank you for your suggestion, we have corrected them.*

L22: "are the largest marine bivalves" and "carbonaceous shell" and "can be used for high-resolution paleoclimate reconstructions".

*Done.*

L47: delete "of".

*Done.*

L48: "physicochemical" is weird in this context – do you want to record environmental signatures encoded within the biocarbonate or do you want to look at physiological variations that may or may not be influences by external factors?

*We have rewrote this expression. Here, we refer to both environmental records by biochemistry ($\delta^{18}O$) and ontogenetic change (e.g. daily increment with dark/light couples) in Tridacna. "physicochemical" has been replaced by "biochemical and ontogenetic".*

L49: "on past climate dynamics" delete "the".

*Done.*

L50-51: I recommend also citing the most recent work on the crossed-lamellar shell architecture of Tridacna see reference: Agbaje, O. B. A., R. Wirth, L. F. G. Morales, K. Shirai, M. Kosnik, T. Watanabe, and D. E. Jacob. "Architecture of crossed-lamellar bivalve shells: the southern giant clam (Tridacna derasa, Röding, 1798)." Royal Society open science 4, no. 9 (2017): 170622.

*Done.*

L54: I doubt that Tridacna lives up to "few centuries" where is the evidence (reference)? This may have been mixed up with Arctica shells or other long-lived bivalves but these are very different from Tridacna!

*We apologies for having made a mistake in the text and have changed the expression to "from several decades to about a hundred year". Some people in China said they had found an about 200 years old Tridacna gigas, but it has not been confirmed by authorities. From the Tridacna gigas we collected, the oldest one had lived about 100 years, most of them are between 30 to 60 years.*

L57: "precipitate" is really a wrong term when talking about shells as it is closely associated with classical crystallisation pathways (i.e. "inorganic" systems). However,

we know for more than over a decade now that shells form by non-classical crystallization pathways via precursor phases (amorphous calcium carbonate and/or vaterite). I am not saying you need to venture into the area of shell biomineralization here but I would strongly argue to find a better word for this text passage. Maybe replacing "precipitate their shells" with simply "grow".

*We accept the referee's suggestion, and have replaced the word "precipitate" into "grow".*

L59-60: What do you mean with "ontogenetic reduction"?

*"ontogenic reduction" refers to the decreasing growth rate with ontogeny. This word was mentioned by K. Welsh (2011). As K. Welsh indicated in his article, the ontogenic reduction in growth of T. gigas does not reduce the reliability with which temperature and $\delta^{18}O_w$ variability can be reconstructed. Climate reconstruction in $\delta^{18}O_{shell}$ don't have an incongruity with temperature and $\delta^{18}O_w$ which might be an obviously declined or increased tendency. We have rephrased this sentence to make this clearer:*
and the reliability in reconstruction between temperature and $\delta^{18}O_w$ variability would not be reduced by the ontogenic reduction in growth of the *Tridacna* $\delta^{18}O$.

L80: "occurring nowadays", however, I think you should try and find a more appropriate word than "nowadays" as this sounds perhaps too casual and please replace throughout manuscript.

*As suggested by the reviewer, we have replaced the word "nowadays" into "recent decades" or "present".*

L83-84: Better: "High-resolution isotopic geochemical data from Tridacna may provide detailed insight into climatic variations of this period."

*Done.*

L117: "give distinct seasonal SST to the Tridacna from the coral reefs" reads clumsy, perhaps change to "provide distinct seasonal SST for Tridacna populating the coral reefs of the Xisha Islands".

*Done*

L123-125: I don't understand "rehandling" do you mean "re-sampling"? I agree with referee 1 that this sentence needs to be rewritten for more clarity. Please change throughout the manuscript.

*Thanks for the suggestion. We agree with you that change the verb "rehandle" into "resample" will be better. The technique we used for the resampling is a cubic spline*

*model in AnalySeries 2.0.8. This method was first applied by Schöne and Fiebig (2009), who used bivalve shells (Arctica islandica) to reconstruct climate. They suggested that 7 points per month would elapse during the core growing season of the shell (i.e., time interval of fastest shell growth covering the seasonal extremes). And only the annual sample number for which equal to or more than seven existed could be used. Therefore, we used 7 points per month.*

L130-131: Perhaps better: "It is excluded that river runoff effects SSS as the Xisha Islands are at a XXX km distance to the continental mainland." Please quantify roughly to provide evidence.

*Done.*

L138-143: I recommend providing a sentence regarding the crossed-lamellar shell architecture of Tridacna see above mentioned reference Agbaje et al. (2017).

*We thank the referee for his/her advice and have added them in the manuscript as follow:* Study in shell architecture showed a crossed lamellar microstructure with a strong fibre texture made the mechanical properties of those bivalve shells more optimized (Agbaje et al., 2017).

L144: when you mention "$^{14}$C AMS" for the first time I recommend providing the full method name in brackets (replace "$^{14}$C AMS" with "$^{14}$C AMS (Accelerator Mass Spectrometry)") for readers that lack this methodological background.

*Done.*

L145: I don't understand the meaning of "conventional" in this sentence – maybe not the right phrase? What is the uncertainty? First or second standard deviation or something else?

*From the modern Tridacna samples we collected in this area, the dating results showed no obvious "reservoir effect" (Liu et al., 2019). Tridacna might exchange its carbon with the atmosphere through photosynthesis. Therefore, we used the atmospheric $^{14}$C yield model to calibrated. We have clarified the details about the radiocarbon date calibration as followed:* The radiocarbon age determination was performed at Institute of Earth Environment of Chinese Academy of Sciences. The $^{14}$C Accelerator Mass Spectrometry data revealed the fossil *Tridacna gigas* age was $3437 \pm 28$ yr BP. Due to no obvious "reservoir effect" in dating results of modern *Tridacna* shells, the atmospheric $^{14}$C yield model was used to calibration. The calibrated date ($2\sigma$) was range from 1783 to 1663 cal BC, with the median date is 1741 cal BC by using the IntCal13 of Radiocarbon Calibration Program CALIB 7.10.

L154: "from adult to childhood" is not the right phrase how about "in a transect from adult to ontogenetically younger shell"?

*Thank you for your advice, we have replaced this phrase.*

L185: "40 dark/light couples (each representing one year)" please explain how dark/light line couples relate to time/tide schedules/seasonality. How much time/which tide pattern does one dark-light line couple stand for?

*We have added them to clarify as follow:*
*From the shell slice section, dark/light line couples (each couple represents one year) can be seen clearly (Fig. 1c, Fig. 3a). Follow the $\delta^{18}O_{A5}$ profiles, those short and dark lines (transparent) corresponding to higher $\delta^{18}O_{A5}$ values, which means Tridacna grew in low temperature (cold seasons such as December to February). In contrast, lower $\delta^{18}O_{A5}$ values lie in the long and light lines (opaque), corresponding to the high temperatures (warm seasons such as March to November).*

L192: Increments are not obvious to me from the image. Especially Fig. 3b is not clear what one should see, perhaps choose a different image with better resolution.

*It's really hard to take a clear picture from Tridacna A5 for the organic matter influence. Those organic matter covered most of increments and make those increments unclear. We had tried our best to find this picture under microscope with obvious increments change. We have retreated the picture contrast and brightness to make them clear as the reviewer's suggestion.*

[Figure]

Figure 3. (a) Dark/light lines consistent with $\delta^{18}O_{A5}$ profiles. Dark and light lines correspond to high $\delta^{18}O$ (cold seasons) and low $\delta^{18}O$ (warm seasons), respectively. The distance between the dash lines represents a year that *Tridacna* grew. Blue line represents the sampling line. (b) Under the microscope, daily increments (a dark coupled with a light increment) grow slower when seasons are cold, but faster when the temperature rises up. (c) Growth rates (line 2 in Fig. 1c) in fossil *Tridacna* A5.

L192-193: "In general, Tridacna A5 grew faster in warm seasons and slower in cold seasons (Fig. 3b)." Where is your evidence for this assumption? I feel you need to back this up as this varies between species and you need to demonstrate to the reader that it is the case for Tridacna. Also, more seasonal information may be needed to achieve this. How long are summers how long are winters? For example: if a reader believes summer and winter are similar in length one could misinterpret short low $\delta^{18}O$ periods may have just been formed quicker (and have thus higher not lower growth rates!). This all needs more explanation and demonstration and is important as you build upon this later in the discussion. Perhaps see other papers I suggest any study by Carré et al as they are very educative in this respect.

*This evidence focuses on Fig. 3b and we added more clarification in section 2.4. As we mentioned above, it's hard to see entirely increments in a year because of the organic matter influence. However, some fragments near the highest $\delta^{18}O_{shell}$ (indicating this period happened in cold season) show the increments change as the $\delta^{18}O_{shell}$ become lower (temperature become higher). From Fig. 3b, the daily increment is about 2.7 μm in low temperature, while as the temperature rises up, the daily increment can reach to 5.7 μm. It happened normally throughout Tridacna's life. Therefore, we have this conclusion that Tridacna grew faster in warm seasons and slower in cold seasons. Meteorological observations reveal that the cold seasons happened from about December to February, the rest of months are relatively suitable for Tridacna to grow fast. But it's hard to distinguish exactly how long is cold and how long is warm through Tridacna's increments. The growth rates influence on $\delta^{18}O_{shell}$ cannot be eliminated. However, we use the resampling method suggested by Schöne and Fiebig (2009), which try to reduce this problem as much as possible.*

L196-197: I don't understand this sentence.

*As suggested by Schöne and Fiebig (2009), the technique we used for the resampling that would elapse during the core growing season of shell (i.e., time interval of fastest shell growth covering the seasonal extremes). Also, before we resample the data, the numbers of annual data change because of different growth rates. To some extent, data resampling makes annual data become comparable.*

L201: Perhaps not everything about Tridacna but $\delta^{18}O$?

*You are right, we have changed this phrase into "oxygen isotopic equilibrium".*

L259: "lived 3700 years ago" delete "in".

*Done.*

L286-287: Better: "Due to a higher sampling density in Tridacna: : :".

*Done.*

L288: "magnified" is the wrong word here.

*Thank you. We have replaced the word into "enlarged".*

L292: "switching" wrong word, replace throughout manuscript.

*Thank you. We have replaced the word into "transition".*

L293-294: This sentence contradicts itself and needs rewording for clarity.

*The seasonality is the range between the lowest temperature and the highest temperature in the text. In order to eliminate the different influence in location, we use the reconstructed $SST_{A5}$ and North Reef SST (from NOAA) to compare and the result shows the seasonality in 3700 years ago had slightly lower. Besides, the transition between cold to warm seasons focuses on the slope of $\delta^{18}O_{shell}$ when temperature change from low to high (or high to low), mainly focus on $1^{st}$ r-month to $2^{nd}$ r-month (or $6^{th}$ r-month to $7^{th}$ r-month). This situation is better to compare between two $\delta^{18}O_{shell}$ (modern and fossil) because the monthly data are not equal to evenly instrumental data. Therefore, we can see in Fig. 4c, the slope of A5 is obviously sharper than YX1, which means the transition between cold to warm seasons was more serious 3700 years ago. In conclusion, we consider that the climate around 3700 years ago had slightly lower seasonality than present, and the transition between cold to warm seasons was more serious.*

L296-299: reads more like an introduction section and is not relevant here (suggest to delete).

*As suggested by both two reviewers, we have removed this section.*

L303: "instrumentation data" is odd.

*Thank you. We have replaced the word into "modern instrumental data".*

L326: "calcite-affected" sounds also a bit odd to me maybe you can find a better term. Why is calcite "bad" in this sense? Why is it a limitation?

*We apologize for this confusion in the text. We have replaced this sentence into "such as those concerning the post-depositional diagenetic alteration between aragonite and calcite". As McGregor and Gagan (2003) indicated in their research, some corals had both aragonite and calcite in their skeleton, the range between them in $\delta^{18}O$ could reach to nearly 3‰. Such alteration should be paid more attention before we use for accurate paleoclimate reconstructions.*

L326-328: Better perhaps: "Analyses of Tridacna species are performed to overcome this limitation by taking advantage of their denser shells, negligible diagenetic alteration, and oxygen isotopic equilibrium with seawater."

*Done.*

L338: unclear.

*Do you mean that there is an unclear about which one we bring 3-month forward? We added this for clarification:*

According to the SST series, the North Reef SST have a 3-month time lag behind the Ninõ 1 + 2 SST (Fig. 8a), and thus we bring 3-month of the North Reef SST forward to eliminate the lag.

Figure 1: It looks like your 5 cm scale bar is too large for the scale in the figure (measuring tape, here 5 cm look smaller). There are some grammar issues in the figure caption.

*We apologize for having made this mistake in the scale bar and have replaced the right one. Figure caption had rewritten as follow:*

[Figure]

Figure 1. (a) Maps of the South China Sea, with the location of the sample study area in the Xisha Islands. (b) Photo of integral *Tridacna* A5, a slice was cut from the red line of integral *Tridacna* A5. (c) Different parts can be seen (hinge, inner layer, and outer layer), the red lines are the sampling lines for δ18O analysis. (d) Meteorological observations in the Xisha Islands from 1994 to 2005: R-monthly average air temperature (AT) and sea surface temperature (SST), the error bars reveal the highest

and the lowest temperature in the month; (e) R-monthly average rainfall and sea surface salinity (SSS) with standard deviation (1σ).

L633: "amplitude" may not be the right word here.

*We have removed this word and rewroted this sentence.*

L635-636: "under the microscope, daily increments grow slower in cold seasons, but faster in warm seasons" – this is not visible from microscope images alone! This needs more explanation! Also, image is not really easy to understand (what should be seen? It's all very blurry).

*As suggested by the reviewer, we have replaced the photo of Fig. 3b and added for more clarification in the 2$^{nd}$ paragraph of section 3.2:*
Furthermore, daily increments (a dark coupled with a light increment) can be seen under the microscope (Fig. 3b). Here, a fragment was chosen where δ$^{18}$O was near highest in a year. This period lied on the cold season which daily increment was about 2.7 μm. When the temperature rose up as warm season began, *Tridacna* grew faster that daily increment could reach to 5.7 μm. This situation occurred throughout *Tridacna*'s life. In general, *Tridacna* A5 grew faster in warm seasons and slower in cold seasons.

***Reference***
Schöne, B. R. and Fiebig, J.: Seasonality in the North Sea during the Allerød and Late Medieval Climate Optimum using bivalve sclerochronology, Int. J. Earth Sci., 98(1), 83–98, doi:10.1007/s00531-008-0363-7, 2009.

McGregor, H. V. and Gagan, M. K.: Diagenesis and geochemistry of Porites corals from Papua New Guinea: Implications for paleoclimate reconstruction, Geochim. Cosmochim. Acta, 67(12), 2147–2156, doi:10.1007/430_2015_174, 2003.

Welsh, K., Elliot, M., Tudhope, A., Ayling, B. and Chappell, J.: Giant bivalves (*Tridacna gigas*) as recorders of ENSO variability, Earth Planet. Sci. Lett., 307(3–4), 266–270, doi:10.1016/j.epsl.2011.05.032, 2011.

Liu, C., Yan, H., Fei, H., Ma, X., Zhang, W. and Shi, G.: Journal of Asian Earth Sciences Temperature seasonality and ENSO variability in the northern South China Sea during the Medieval Climate Anomaly interval derived from the Sr / Ca ratios of *Tridacna* shell, J. Asian Earth Sci., 180(June), 1-9, doi:10.1016/j.jseaes.2019.103880, 2019.

---

## Referee Report (RR1)

General comments:

I can see from their comments and corrections in the manuscript that Hu et al. have made great efforts to accommodate all reviewer suggestions and this has greatly improved the manuscript. However, I have three objectives that I feel need to be addressed before this manuscript can move forward:

Firstly, the writing has still many language issues (especially the new yellow text passages) and I would like to urge the authors to seek some editorial support either through the journal or within their professional network. I believe it is not the reviewer's job to correct the entire grammar of a manuscript but to evaluate and support the intellectual achievement. I will do my best to provide some additional language help but need to emphasise here that this is just scratching the surface. Also, missing line numbers in the most recent version make it very difficult to provide specific comments.

Secondly, I am still not convinced that fastest growth rates correlate with warm seasons. It could be the case but (in theory) it's a 50/50 chance so you need to demonstrate this clearly. I understand that it may be challenging to visualise them from the fossil shell but why don't you use your modern shell for this? I assume you know when the shell was collected and sacrificed (summer or winter) you could prepare a microscope image showing the inner growth front and check if a dark or bright (opaque) line was formed most recently. There shouldn't be any interfering organic matter in this sample. I believe this extra effort could go a long way and greatly support your argumentation with direct evidence.

Lastly, I believe the arrow indicating the direction of growth in Figure 3 has a wrong orientation and it is missing in Figure 1 altogether. Such shell orientation "landmarks" are important and need to be presented for the reader to understand the shell geometry. A growth direction arrow always refers to the dorso-ventral shell extension and, thus, will never be perpedicular to the growth lines of an inner shell layer (as in this case for the inner layer of *Tridacna* here). It will be roughly parallel to these growth lines – as the inner surface of the shell is not depicted in this image there are 2 options – it could be pointing up or downwards (as indicated by my red arrows). Authors need to check where on the section they took the image and re-draw the arrow accordingly. Alternativly for Figure 3 authors could consider leaving the arrow as it is but modify the text to read "local direction of growth", then this arrow would indeed indicate the orientation of the inner shell layer (instead of the dorso-ventral growth direction, see Otter et al 2019 for more explanation on shell growth directions).

[Figure]

Growth direction (defined as dorso-ventral shell extension)

I have also inserted direction of growth arrows in Figure 1 for the authors as they should consider adding it to their figure (smaller and black of course I exaggerate here just for the purpose of clarity). For further information on the difference in general and local shell growth directions I recommend reading and citing Otter et al. "Insights into architecture, growth dynamics, and biomineralization from pulsed Sr-labelled Katelysia rhytiphora shells (Mollusca, Bivalvia)." Biogeosciences 16.17 (2019): 3439-3455, especially chapter 3.6.

[Figure]

I hope this second round of suggestions does not discourage the authors and hope they can view these comments as a last "fine tuning" of their manuscript and hope they will undertake this last effort. I look forwards to seeing the final published version of this study!

Specific comments:

**Abstract:**

omit "in the tropical ocean" as Tridacna is the largest bivalve in general.

**Introduction:**

delete "and ontogenetic" as this reads confusing (they do not help to understand past climate it is just an intrinsic growth pattern of the animals).

"Bivalves, which are considered to be high-resolution records, can give us more precisely environmental variation details." Change to "Bivalve shells" as the soft tissue is not used.

"Studies in bivalve mollusk specimen (Arctica islandica) oxygen isotopes showed a different seasonal temperature change compared between the Little Ice Age and the present" it is not clear what you mean with this sentence and why you give this information. These literature examples need to be better embedded within your story. Why is this relevant?

"Previous studies indicated that Tridacna species grow their shells with the oxygen isotopic ($\delta18O$) equilibrium with seawater" change to "Previous studies indicated that Tridacna grow their shells in oxygen isotopic ($\delta18O$) equilibrium with the surrounding seawater" and better end the sentence after the provided references and modify beginning of next sentence.

Check for past and present tenses: "…studies on the mid to late Holocene ENSO evolution yield controversial findings" and "Coral records show" "fossil mollusk shells suggest" Ask yourself do they still show? Is it still the case? Then use present tense when speaking about earlier findings.

**Materials and methods:**

"Due to suggested in the method (Schöne and Fiebig, 2009) and the yearly minimum number of $\delta18O_{YX1}$ was seven, the time-scale of modern Tridacna YX1 is resampling into seven points/yr, which indicates a resampled month (r-month) represents 1.7 actual month. All meteorological observations and $\delta18O_{shell}$ are using this method to resample the time-scale" needs to be rewritten for clarity (grammar).

"Study in shell architecture showed a crossed lamellar microstructure with a strong fibre texture made the mechanical properties of those bivalve shells more optimized (Agbaje et al., 2017)" perhaps change to "A recent study investigating the architecture of Tridacna shells shows a crossed lamellar microstructure with a strong fibre texture with optimised mechanical performance (Agbaje et al., 2017)."

…performed at the Institute of Earth Environment of the Chinese…" and "the fossil Tridacna gigas age is 3437…"

"…the standards and samples had reproducibilities (1σ) of better…"

The last paragraph of the methods section needs more extensive language editing.

**Results**

"From the shell section" (delete slice)

"…which suggests Tridacna grew in low temperature (potentially from December to February)." And "In contrast, lower $\delta18O_{A5}$ values lie in the wider bright, opaque lines…" careful with the word "lighter" this refers to density and weight – what you mean is "brighter" – check and edit throughout the manuscript.

"Furthermore, daily increments visible as pairs of dark and bright increments can be seen…"

"This period fell into the cold season with daily increments of about 2.7 μm width." You mean one dark and bright pair?

**Discussion:**

"R-monthly mean values are used to compare for they are minimizing the influence of extreme events." It is not clear what you mean – what do you compare?

Perhaps better: "As ENSO is the strongest contributor to global interannual climate variations a better understanding of its fundamental properties will allow us to better unravel past climate change episodes and to make more accurate predictions for the future"

---

## Author Response (AR2)

*Anonymous Reviewer #1:*
*We would like to thank this reviewer for her/his professional comments on our manuscript. Please find our detailed answers to each comment below. The reviewer comments are in normal black script, our answers are in blue italics and the revised texts are in blue normal script.*

The data presented is of high quality and the method is robust.

A lower frequency but larger amplitude of ENSO events is found 3700 years ago, which is very interesting.

However, the authors find a larger variability (estimated by the standard deviation) of temperature anomalies in the fossil record and yet conclude (including in the title) for a reduced activity of ENSO at that time. The "activity" of ENSO is a widely used but poorly defined term that includes both frequency and amplitude. The standard deviation of interannual SST anomalies also integrates frequency and amplitude of events and is therefore usely used as an estimate of ENSO activity.

This means that the conclusion should not point to a reduced activity of ENSO but to an increased activity, or , more precisely, to an increased ENSO-related variability in this region.

*Thank you for your professional suggestion. We have corrected the "reduced ENSO activity" into "reduced ENSO frequency", and use "ENSO-related variability" to indicate the standard deviation of interannual SST anomalies. The conclusion has also corrected.*

The contrast with reconstructions from the eastern Pacific (Koutavas et al., 2012, Carré et al., 2014) may point to interesting changes in ENSO flavours.

*It is really interesting to find an opposite conclusion on the other side of the Pacific on the same time-scale. We have added sentences to discuss this contrast in the last part of 4.4:*
As a result, the enhanced climate variability 3700 years ago probably indicates increased ENSO-related variability in this region. This conclusion contradicts data from samples (deep-sea sediments and fossil mollusk shells) collected in the eastern tropical Pacific at the same time period (Koutavas et al., 2012, Carré et al., 2014). More data should be analyzed from long, successive time periods to understand more about the dynamics of ENSO on a large scale.

The paleo-ENSO litterature seems not completely understood. Some references are sometimes not properly cited. Tudhope et al. 2001, for instance is once cited to support decreased variability at 3.7ka and somewhere else for the opposite.

*We apologize for this mistake. We have checked and corrected them in rivision.*
.
The english still needs to be improved.

*We have asked an English language editor for help to improved our manuscript this time, who have worked in CP copy-editing services before, and hope it's better than before for you to comprehend our study.*

The original isotopic data of modern and fossil shells should be included as supplementary material, or available in a public repository such as Pangaea.

*We have added the supplementary material of both shells. The isotopic data of Modern shell are in Table S1, the data of fossil shell are in Table S2.*
*We would like to thank this reviewer for her/his comments on our manuscript. Please find our detailed answers to each comment below. The reviewer comments are in normal black script, our answers are in blue italics and the revised texts are in blue normal script.*

General comments:

I can see from their comments and corrections in the manuscript that Hu et al. have made great efforts to accommodate all reviewer suggestions and this has greatly improved the manuscript. However, I have three objectives that I feel need to be addressed before this manuscript can move forward:

Firstly, the writing has still many language issues (especially the new yellow text passages) and I would like to urge the authors to seek some editorial support either through the journal or within their professional network. I believe it is not the reviewer's job to correct the entire grammar of a manuscript but to evaluate and support the intellectual achievement. I will do my best to provide some additional language help but need to emphasise here that this is just scratching the surface. Also, missing line numbers in the most recent version make it very difficult to provide specific comments.

*Thank you for your additional language help to us. We have asked an English language editor for help to improve our manuscript once again, and hope it's better than before for you to comprehend our study. Also, line numbers are added to the manuscript.*

Secondly, I am still not convinced that fastest growth rates correlate with warm seasons. It could be the case but (in theory) it's a 50/50 chance so you need to demonstrate this clearly. I understand that it may be challenging to visualise them from the fossil shell but why don't you use your modern shell for this? I assume you know when the shell was collected and sacrificed (summer or winter) you could prepare a microscope image showing the inner growth front and check if a dark or bright (opaque) line was formed most recently. There shouldn't be any interfering organic matter in this sample. I believe this extra effort could go a long way and greatly support your argumentation with direct evidence.

*Your suggestion about using the modern shell is really good to support our argumentation. In fact, we do use a modern Tridacna, collected in Xisha Islands with an accurate daily time-scale, to analyze the growth rate change under daily increments by using laser scanning confocal microscope. The result shows it is true that Tridacna from Xisha Islands grows fast in warm while relative slow in cold temperature. However, this paper is accepted by PNAS and will be published soon. We apologize here that we cannot cite right now.*

Lastly, I believe the arrow indicating the direction of growth in Figure 3 has a wrong orientation and it is missing in Figure 1 altogether. Such shell orientation "landmarks" are important and need to be presented for the reader to understand the shell geometry. A growth direction arrow always refers to the dorso-ventral shell extension and, thus, will never be perpedicular to the growth lines of an inner shell layer (as in this case for the inner layer of *Tridacna* here). It will be roughly parallel to these growth lines – as the inner surface of the shell is not depicted in this image there are 2 options

– it could be pointing up or downwards (as indicated by my red arrows). Authors need to check where on the section they took the image and re-draw the arrow accordingly. Alternativly for Figure 3 authors could consider leaving the arrow as it is but modify the text to read "local direction of growth", then this arrow would indeed indicate the orientation of the inner shell layer (instead of the dorso-ventral growth direction, see Otter et al 2019 for more explanation on shell growth directions).

[Figure]

Growth direction (defined as dorso-ventral shell extension)

I have also inserted direction of growth arrows in Figure 1 for the authors as they should consider adding it to their figure (smaller and black of course I exagerate here just for the purpose of clarity). For further information on the difference in general and local shell growth directions I recommend reading and citing Otter et al. "Insights into architecture, growth dynamics, and biomineralization from pulsed Sr-labelled Katelysia rhytiphora shells (Mollusca, Bivalvia)." Biogeosciences 16.17 (2019): 3439-3455, especially chapter 3.6.

[Figure]

*Thank you for your professional advice. We have added more details of shell geometry in our figures refer to Gannon et al. (2017).*

*Besides, we apologize for making confusion about the word "growth direction". We know the Tridacna grow its whole shell from that you demonstrate, but what we want to express is the direction that Tridacna precipitates its shell from young to old. According to Gannon et al. (2017), who study the biomineralization of Indo-Pacific giant clam Tridacna gigas, indicate the mineralization of the inner and outer layers is independent from each other and associated with the activity of specific mantles. From Fig. 1 in their*

*research, the layers are different between inner and outer. We acquire our sample of inner layers from the growth of young to old, and each sample is parallel with the layer. We have removed the word "growth direction" into the word "growth (young to old)" to avoid confusion. Also, we have added the description of the shell in 2.2.*

[Figure]

Figure 1

[Figure]

Figure 2

[Figure]

Figure 3

I hope this second round of suggestions does not discourage the authors and hope they can view these comments as a last "fine tuning" of their manuscript and hope they will undertake this last effort. I look forwards to seeing the final published version of this study!

*We appreciate your careful and detailed suggestions for our manuscript. We will do our best to make it more professional! Thank you again!*

Specific comments:

**Abstract:**

omit "in the tropical ocean" as Tridacna is the largest bivalve in general.

*Done.*

**Introduction:**

delete "and ontogenetic" as this reads confusing (they do not help to understand past climate it is just an intrinsic growth pattern of the animals).

*Done.*

"Bivalves, which are considered to be high-resolution records, can give us more precisely environmental variation details." Change to "Bivalve shells" as the soft tissue is not used.

*Done.*

"Studies in bivalve mollusk specimen (Arctica islandica) oxygen isotopes showed a different seasonal temperature change compared between the Little Ice Age and the present" it is not clear what you mean with this sentence and why you give this information. These literature examples need to be better embedded within your story. Why is this relevant?

*Another referee suggests us to add some researches about other bivalves. We have rewrote*

*this sentence and changed it into the forth paragraph of introduction. We want to emphasize the high-resolution isotopic data in other bivalves have been paid attention, then indicate that Tridacna can provide high-resolution details as well.*

"Previous studies indicated that Tridacna species grow their shells with the oxygen isotopic ($\delta^{18}O$) equilibrium with seawater" change to "Previous studies indicated that Tridacna grow their shells in oxygen isotopic ($\delta^{18}O$) equilibrium with the surrounding seawater" and better end the sentence after the provided references and modify beginning of next sentence.

*We have corrected those sentences.*

Check for past and present tenses: "…studies on the mid to late Holocene ENSO evolution yield controversial findings" and "Coral records show" "fossil mollusk shells suggest" Ask yourself do they still show? Is it still the case? Then use present tense when speaking about earlier findings.

*Thank you! We have corrected them.*

**Materials and methods:**

"Due to suggested in the method (Schöne and Fiebig, 2009) and the yearly minimum number of $\delta^{18}O_{YX1}$ was seven, the time-scale of modern Tridacna YX1 is resampling into seven points/yr, which indicates a resampled month (r-month) represents 1.7 actual month. All meteorological observations and $\delta^{18}O_{shell}$ are using this method to resample the time-scale" needs to be rewritten for clarity (grammar).

*We have rewrote this paragraph into:*

*In order to compare geochemical analyses with monthly environmental data, isotopic records and meteorological observations were resampled according to the method suggested by Schöne and Fiebig (2009). They used bivalve shells (Arctica islandica) to reconstruct the climate, examining seven points per year would minimize the influence of*

different growth rates throughout the year; meanwhile, only the annual sample number for which equal to or more than seven existed could be used. As the yearly minimum number of $\delta^{18}O_{YX1}$ is seven, 1 year includes 7 resampled months (r-months).

"Study in shell architecture showed a crossed lamellar microstructure with a strong fibre texture made the mechanical properties of those bivalve shells more optimized (Agbaje et al., 2017)" perhaps change to "A recent study investigating the architecture of Tridacna shells shows a crossed lamellar microstructure with a strong fibre texture with optimised mechanical performance (Agbaje et al., 2017)."

*Done.*

…performed at the Institute of Earth Environment of the Chinese…" and "the fossil Tridacna gigas age is 3437…"

*Done.*

"…the standards and samples had reproducibilities (1σ) of better…"

*Done.*

The last paragraph of the methods section needs more extensive language editing.

*We have polished this paragraph and hope it will be better for you to comprehend.*

**Results**

"From the shell section" (delete slice)

*Done.*

"…which suggests Tridacna grew in low temperature (potentially from December to February)." And "In contrast, lower $\delta^{18}O_{A5}$ values lie in the wider bright, opaque lines…" careful with the word "lighter" this refers to density and weight – what you mean is "brighter" – check and edit throughout the manuscript.

*Thank you for your suggestion. We have changed this word throughout the manuscript.*

"Furthermore, daily increments visible as pairs of dark and bright increments can be seen…"

*Done.*

"This period fell into the cold season with daily increments of about 2.7 μm width." You mean one dark and bright pair?

*We apologize for wrong numbers here. The "2.7" and "5.7" are the number before we retreated the picture Fig. 3b. After the retreatment, a new clearer daily increment was measured. The correct phrases should be:*

This period fell during the cold season, when the daily growth increment was about 4.5 μm. When the temperature rose as warm season began, Tridacna grew faster, with daily growth increment reaching up to 8 μm.

**Discussion:**

"R-monthly mean values are used to compare for they are minimizing the influence of extreme events." It is not clear what you mean – what do you compare?

[revised manuscript text omitted]